# Procedural Synthesis of Synthesizable Molecules

**Michael Sun[1], Alston Lo[1], Minghao Guo[1], Jie Chen[2], Connor Coley[3], Wojciech Matusik[1]**
[1]MIT CSAIL    [2]MIT-IBM Watson AI Lab, IBM Research    [3]MIT Chemical Engineering
{msun415,alston,wojciech}@csail.mit.edu,    chenjie@us.ibm.com,
ccoley@mit.edu

## Abstract

Designing synthetically accessible molecules and recommending analogs to unsynthesizable molecules are important problems for accelerating molecular discovery. We reconceptualize both problems using ideas from program synthesis. Drawing inspiration from syntax-guided synthesis approaches, we decouple the syntactic skeleton from the semantics of a synthetic tree to create a bi-level framework for reasoning about the combinatorial space of synthesis pathways. Given a molecule we aim to generate analogs for, we iteratively refine its skeletal characteristics via Markov Chain Monte Carlo simulations over the space of syntactic skeletons. Given a black-box oracle to optimize, we formulate a joint design space over syntactic templates and molecular descriptors and introduce evolutionary algorithms that optimize both syntactic and semantic dimensions synergistically. Our key insight is that once the syntactic skeleton is set, we can amortize over the search complexity of deriving the program's semantics by training policies to fully utilize the fixed horizon Markov decision process imposed by the syntactic template. We demonstrate performance advantages of our bi-level framework for synthesizable analog generation and synthesizable molecule design. Notably, our approach offers the user explicit control over the resources required to perform synthesis and biases the design space towards simpler solutions, which is particularly promising for autonomous synthesis platforms. Supporting code is at https://github.com/shiningsunnyday/SynthesisNet.

## 1 Introduction

The discovery of new molecular entities is central to advancements in fields such as pharmaceuticals (Zhavoronkov et al., 2019; Lyu et al., 2019), materials science (Hachmann et al., 2011; Janet et al., 2020), and environmental engineering (Zimmerman et al., 2020; Yao et al., 2021). Traditional make-design-test workflows for molecular design typically rely on labor-intensive methods that involve a high degree of trial and error (Sanchez-Lengeling & Aspuru-Guzik, 2018). Systematic and data-efficient approaches that minimize costly experimental trials are the key to accelerating these processes (Coley et al., 2020a;b; Gao et al., 2022). In recent years, a large number of molecular generative models has been proposed (De Cao & Kipf, 2018; Ma et al., 2018; Simonovsky & Komodakis, 2018; You et al., 2018; Li et al., 2018; Samanta et al., 2020; Liu et al., 2018; Jin et al., 2018; 2020; Guo et al., 2022; Sun et al., 2024). However, few of their outputs are feasible to make and proceed to experimental testing due to their lack of consideration for synthesizability (Gao & Coley, 2020). This has motivated recent methods that integrate design and synthesis into a single workflow, aiming to optimize both processes simultaneously (Button et al., 2019; Bradshaw et al., 2019; 2020; Gao et al., 2021; Swanson et al., 2024) which significantly closes the gap between the design and make steps, reducing cycle time significantly (Koscher et al., 2023; Volkamer et al., 2023; McCorkindale, 2023). This development is spurred by the curation of a small but robust collection of expert reaction templates that are inspired by real-world reactions and defined in close collaboration with chemists Hartenfeller et al. (2011). This workflow facilitates de novo applications by imposing synthetic accessibility by design, recasting molecular design as navigating the space of possible synthetic procedures over a set of building blocks and forward reaction steps, as defined in Vinkers

et al. (2003). However, these methods still face computational challenges, particularly in navigating the combinatorial explosion of potential synthetic procedures (Smith, 1997).

Inspired by techniques in program synthesis, particularly syntax-guided synthesis (Alur et al., 2013), our method decouples the syntactical template of a synthetic procedure (the *skeleton*) from their chemical semantics (the *substance*). This bifurcation allows for a more granular optimization process, wherein the syntactical and semantic aspects of reaction pathways can be optimized independently yet synergistically. Our methodology employs a bi-level optimization strategy. The upper level optimizes the syntactic template of the synthetic pathway, and the lower level fine-tunes the molecular descriptors within that given structural framework. This dual-layered approach is facilitated by a surrogate policy, implemented by a graph neural network, that propagates embeddings top-down following the topological order of the syntactical skeleton. This ensures that each step in the synthetic pathway is optimized in context, respecting the overarching structural strategy while refining the molecular details. We address the combinatorial explosion in the number of programs using tailored strategies for fixed horizon Markov decision process (MDP) environments. This algorithm amortizes the complexity of the search space through predictive modeling and simulation of Markov Chain Monte Carlo (MCMC) processes (Metropolis et al., 1953; Hastings, 1970; Gilks et al., 1995), focusing on the generation and evaluation of syntactical skeletons. By leveraging the inductive biases from retrosynthetic analysis without resorting to retrosynthesis search, our approach combines accuracy and efficiency in "synthesizing" synthetic pathways. In summary, the contributions of this work are:

1. We reconceptualize molecule design and synthesis as a conditional program synthesis problem, establishing common terminology for bridging the two fields.

2. We propose a bi-level framework that decouples the syntactical skeleton of a synthetic tree from the program semantics. Then, we introduce amortized algorithms within our framework for the tasks of synthesizable analog recommendation and synthesizable molecule design.

3. We demonstrate improvements across multiple dimensions of performance for both tasks, and include in-depth visualizations and analyses for understanding the source of our method's efficacy as well as its limitations.

## 2 RELATED WORKS

### 2.1 SYNTHESIZABLE ANALOG GENERATION

The problem of synthesizable analog generation aims to find molecules close to the target molecule that are *synthesizable*. Closely related but distinct from this problem is computer-assisted retrosynthetic analysis, which has developed through the decades (Corey et al., 1985) in tandem with computers, and is now known as retrosynthetic planning due to its resemblance to more classical tests of AI based around planning. As retrosynthetic planning is done by working backwards (top-down), partial success is not straightforward to define. In other domains, procedural modeling is a bottom-up generation process that generates analogs by design (Merrell & Manocha, 2010; Merrell et al., 2011; Müller et al., 2006; Merrell, 2023). Thus, synthesizable analog generation warrants more specialized methods. Prior works such as Dolfus et al. (2022); Levin et al. (2023) address this by performing alterations of existing retrosynthesis routes, but this constrained approach severely limits the diversity of analogs. Instead, we neither start from a search route nor constrain the search route, but instead, extract analogs via the iterative refinement of the program's syntactical skeleton with inner loop decoding of the program semantics in a bi-level setup. We implement the iterative refinement phase using an MCMC sampler with a stationary distribution governed by similarity to the target being conditioned on. This is a common technique used to search over procedural models of buildings, shapes, and furniture arrangements (Merrell et al., 2011; Talton et al., 2011; Yu et al., 2011), and we showcase its efficacy for the new application domain of molecules.

### 2.2 SYNTHESIZABLE MOLECULE DESIGN

The problem of synthesizable molecule design is to design the synthetic pathway, or *program*, whose output molecule optimizes some property oracle function. Note that unlike generic molecular optimization approaches, the design space is reformulated to guarantee synthesizability by construction. The early works to follow this formulation (Vinkers et al., 2003; Hartenfeller et al., 2011; Button

et al., 2019) use machine learning to assemble molecules by iteratively selecting building blocks and virtual reaction templates to enumerate a library, with recent works such as Swanson et al. (2024) obtaining experimental validation. The key computational challenge these methods must address is how to best navigate the combinatorial search space of synthetic pathways. Prior efforts using bottom-up generation (Bradshaw et al., 2019; 2020) probabilistically model synthetic trees as sequences of actions. These adopt an encoder-decoder approach to map between a continuous latent space and a complex combinatorial space. This results in low reconstruction accuracy and hinders the method on the task of conditional generation. Instead, SynNet (Gao et al., 2021) admits a unified framework for solving analog generation and molecule design by formulating the problem as an infinite-horizon MDP and doing amortized tree generation conditioned on Morgan fingerprints. We show improvements to both tasks through our novel formulation.

## 2.3 PROGRAM SYNTHESIS

Program synthesis is the problem of synthesizing a function $f$ from a set of primitives and operators to meet some correctness specification. A program synthesis problem entails: (1) a background theory $\mathsf{T}$ that is the vocabulary for constructing formulas, (2) a correctness specification, i.e., a logical formula involving the output of $f$ and $\mathsf{T}$, and (3) a set of expressions $L$ that $f$ can take on, described by a context-free grammar $G_L$. In molecular synthesis, we can formulate $\mathsf{T}$ as containing operators for chemical reactions, constants for reagents, molecular graph isomorphism checking, and so forth. The correctness specification for finding a synthesis route for a molecule $M$ is simply $f(\mathcal{B}) = M$, where $\mathcal{B}$ is a set of building blocks, and we seek to find an implementation $f$ from $L$ to meet the specification. A coarser specification is to match the molecule's fingerprint, $\mathrm{FP}(f(\mathcal{B})) = \mathrm{FP}(M)$, and as shown by Gao et al. (2021), this relaxed formulation enables both analog generation and fingerprint-based molecule optimization. Our key innovation takes inspiration from the line of work surrounding syntax-guided synthesis (Alur et al., 2013; Schkufza et al., 2013). Syntax guidance explicitly constrains $G_L$, which reduces the set of implementations $f$ can take on (Alur et al., 2013) and facilitates more accurate amortized inference. Further discussion on the connections between program synthesis and molecular synthesis and the similarities to the literature of retrosynthesis are given in App. D.

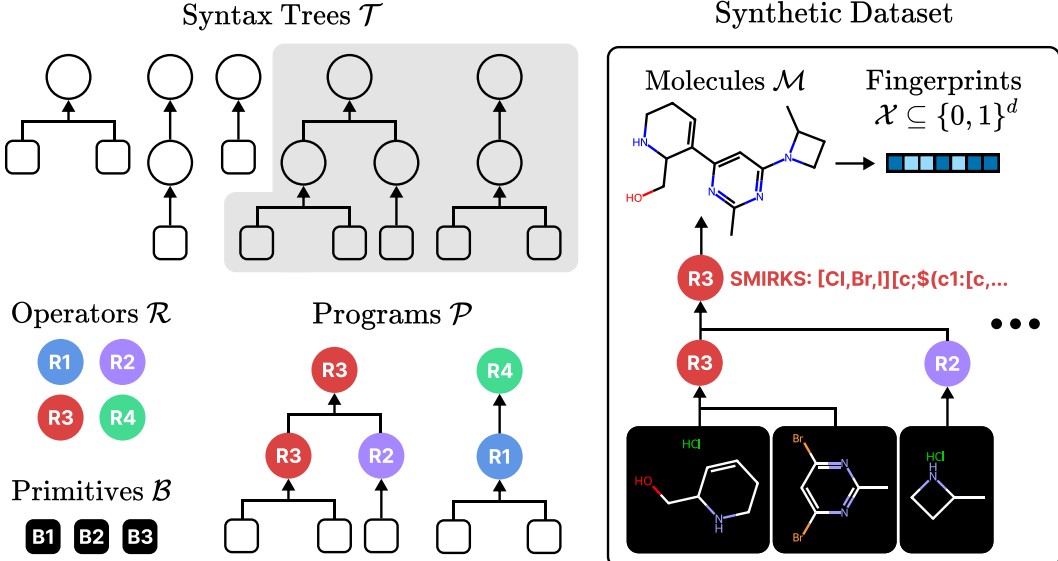

Figure 1: Program synthesis terminology for modeling synthesis pathways.

## 3 METHODOLOGY

### 3.1 PROBLEM DEFINITION

For clarity, we recapitulate the problems of interest discussed in Section 2:

**Synthesizable analog generation** is the inverse task of inferring the program and inputs that best reconstructs a target molecule. Denoting the set of reactions as $\mathcal{R}$ and the set of building blocks as $\mathcal{B}$, we obtain a compact yet expressive design space $\mathcal{P}$: all non-trivial, attributed binary trees where each internal node corresponds to a reaction. Drawing parallels to the program synthesis literature, we call $\mathcal{P}$ the *program* space. The problem can now be formalized: given a space of molecules $\mathcal{M}$, learn a mapping $F\colon \mathcal{M} \to \mathcal{P} \times \mathcal{B}^*$, $M \mapsto (P, B)$ such that $B$ can be assigned to the leaf nodes of $P$ and running the reaction(s) in $P$ in a bottom-up manner (by recursively feeding the product of a node's reaction to its parent) produces a molecule $M_*$ with minimal "distance" to $M$. Using program execution notation, the objective is stated as: $\arg\min_{(P,B)\in\mathcal{P}\times\mathcal{B}^*} \mathrm{dist}(P(B), M)$.

**Synthesizable molecule design** is the forward task of finding the program and inputs whose output optimizes a property oracle function. The oracle can represent property predictors, simulation, experimental validation, etc. but are black-box in nature. The objective is $\arg\min_{(P,B)} \mathrm{Oracle}(P(B))$.

Moving forwards, we identify $(P, B)$ with its output $M_*$ when it simplifies notation. Stripping the semantics from a program leaves behind a syntactic skeleton, which lies in the space $\mathcal{T}$ of non-trivial binary trees. Figure 1 illustrates the discussed terminologies[1].

## 3.2 Solution Overview

Herein, we use expert-defined reaction templates, a popular abstraction codifying deterministic graph transformation patterns in the SMIRKS formal language. SMIRKS assumes the reactants are ordered for defining how atoms and bonds are transformed from reactants to products. Since templates are designed to encompass the most common and robust chemical transformations, ours are restricted to uni-molecular (e.g., isomerization, ring closure, or fragmentation reactions) or bi-molecular (e.g., additions, substitutions, or coupling reactions) reactions. In practice, we featurize molecules using Morgan fingerprints with radius 2 and $d = 2048$ bits, which is a common molecular representation in both predictive and design tasks. This means that $F$ is now technically a map over fingerprint space $\mathcal{X} \subseteq \{0, 1\}^d$. It is then natural to use the Tanimoto distance between fingerprints as our notion of molecular distance.

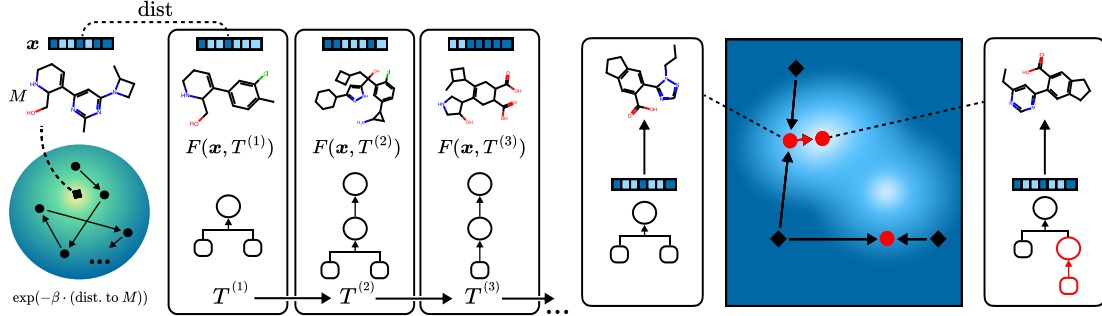

Figure 2: (Left) Our Metropolis-Hastings algorithm in Section 3.3 iteratively refines the syntax tree skeleton towards the stationary distribution which is proportional to the inverse distance to our target molecule $M$. (Right) Our genetic algorithm over the joint design space $\mathcal{X} \times \mathcal{T}$ in Section 3.4 combines the strategies of semantic crossover ($\rightarrow$) and syntactical mutation ($\rightarrow$) to encourage both global improvement and local exploration.

## 3.3 Bilevel Syntax-Semantic Synthesizable Analog Generation

Given a molecule $M \in \mathcal{M}$, we aim to find a program and inputs $(P, B)$ whose output $M_*$ is most similar to $M$. Suppose first that the syntax $T$ of $P$ was given. Then, we would use two learned policies $\pi_{\mathcal{R}}$ and $\pi_{\mathcal{B}}$ that iteratively attribute reactions and building blocks to the nodes of $T$ in topological order (Section 3.3.2). Recalling Section 2.3, this can be seen as parameterizing the derivation process of a context-free grammar $G_{\mathcal{P}}$. We give further details in App. E and discuss how these derivations are constrained through syntax.

---

[1]Please refer to Figure 3 of Gao et al. (2021) for example chemical illustrations of the two tasks.

The application of our policies essentially specifies a syntax-conditioned program synthesis map $F \colon \mathcal{M} \times \mathcal{T} \to \mathcal{P} \times \mathcal{B}^*$. However, $T$ is not known during inference, so to remove this dependence, we consider two further strategies. One is to learn a classifier $\tau \colon \mathcal{M} \to \mathcal{T}$ to predict the most likely syntax tree of the program that produces $M$. We implement $\tau$ with a multi-layer perceptron (MLP) trained under a standard classification task.[2] However, a single prediction may be inadequate, since there can be multiple skeletons corresponding to the same molecule. Instead, our second strategy uses a bi-level setup, where the outer loop explores syntax space $\mathcal{T}$ through invocations of the inner loop $F$, which explores the program's semantics. This approach is further made efficient by amortizing the training of the inner loop.

### 3.3.1 Outer loop: Syntax Tree Structure Optimization

We simulate a Markov process on $\mathcal{T}$ for discovering skeletons whose decoding will maximize the similarity between $M$ and the program's output (Figure 2 (Left)). The details for how we bootstrap and apply $\mathcal{T}$ is in App. A. We adopt the Metropolis-Hastings algorithm with proposal distribution $q(T \mid T_0) \propto \exp(-\lambda\, d_{\mathcal{T}}(T, T_0))$ and scoring function $f(T) \propto \exp(-\beta\, d_{\mathcal{M}}(M, F(M, T)))$, where $\lambda$ and $\beta$ are parameters that trade-off exploration with exploitation, $d_{\mathcal{T}}$ is the tree edit distance, and $d_{\mathcal{M}}$ is the Tanimoto distance. In other words, we use the inner loop to score candidates in $\mathcal{T}$.

### 3.3.2 Inner loop: Inference of Tree Semantics

We now formulate syntax-conditioned derivations under $G_{\mathcal{P}}$ as a finite-horizon MDP.

**State space:** To bridge $\mathcal{T}$ and $\mathcal{P} \times \mathcal{B}^*$, we introduce an intermediate state space of partial programs $\partial\mathcal{P}$ consisting of all possible partial programs arising from the following modifications to any syntax tree $T \in \mathcal{T}$: (1) prepend a new root node of $T$, and attribute it with a fingerprint from $\mathcal{X}$, or (2) attribute some internal (resp. leaf) nodes of $T$ with elements of $\mathcal{R}$ (resp. $\mathcal{B}$). We further require that if a node in $T$ is filled, then so is its parent. Intuitively, $\partial\mathcal{P}$ comprises all partially filled-in trees in $\mathcal{T}$ obeying topological order (with an added root node attached to a molecular fingerprint).

**Action space:** At a state $S \in \partial\mathcal{P}$, the actions are to attribute any frontier node, i.e., unfilled nodes whose parents are filled, with an item from $\mathcal{R}$ (resp. $\mathcal{B}$) if the node is internal (resp. a leaf).

**Policy network:** We parameterize the policies $\pi_{\mathcal{R}} \colon \partial\mathcal{P} \to \mathcal{R}^*$ and $\pi_{\mathcal{B}} \colon \partial\mathcal{P} \to \mathcal{B}^*$ with separate graph neural networks. Given a partial program $S$ as input, the former predicts the reactions that should be attributed to internal frontier nodes as an $|\mathcal{R}|$-label classification problem, while the latter does so for building blocks and leaf nodes. However, $|\mathcal{B}| \gg |\mathcal{R}|$ is very large, so $\pi_{\mathcal{B}}$ instead predicts a 256-dimensional embedding at each nodes which implicitly specifies the building block whose 256-bit Morgan fingerprint is closest.

**Training:** We train both policies using supervised policy learning. Key to this approach is the dataset used for training constructed using Algorithm 1. See App. F for further details.

---

**Algorithm 1** Construction of training dataset.

---

**Require:** A synthetic dataset $\mathcal{D}_0 \subseteq \mathcal{P} \times \mathcal{B}^*$ of programs (Section 4.1.1).
 1: $\mathcal{D} \leftarrow \varnothing$
 2: **for** each $(P, B) \in \mathcal{D}_0$ **do**
 3:     Turn $(P, B)$ into a fully-filled program $T \in \partial\mathcal{P}$ whose root is attributed with $\mathrm{FP}(P, B)$.
 4:     **for** each $\Lambda \in 2^T$ containing the root and closed under $\mathrm{parent}(\cdot)$ **do**
 5:         $\mathrm{Frontier}(\Lambda) \leftarrow \{i \in T \mid i \notin \Lambda \text{ and } \mathrm{parent}(i) \in \Lambda\}$
 6:         Populate node features $\boldsymbol{H}$ and labels $\boldsymbol{Y}$ based on $P$ and $B$ (App. F)
 7:         **for** $i \in T - \Lambda$ **do**
 8:             Mask the feature in $\boldsymbol{H}$ corresponding to node $i$
 9:         **for** $i \in T - \mathrm{Frontier}(\Lambda)$ **do**
10:             Mask the label in $\boldsymbol{Y}$ corresponding to node $i$
11:         $\mathcal{D} \leftarrow \mathcal{D} \cup \{(T, \boldsymbol{H}, \boldsymbol{Y})\}$
    **return** $\mathcal{D}$

---

[2]In practice, we restrict to a finite subset of $\mathcal{T}$ by imposing a maximum number of reaction nodes.

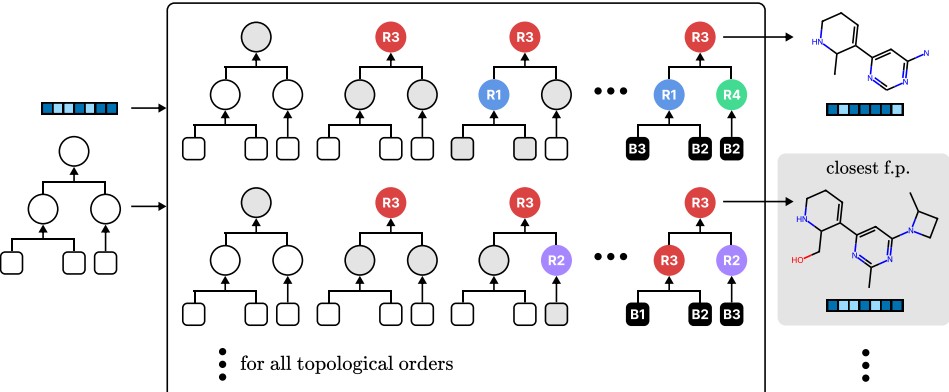

Figure 3: Illustration of our decoding scheme $F$: (Left) The input is a Morgan fingerprint $\boldsymbol{x}$ and syntax skeleton $T$; (Middle) Decode once for every topological ordering of the tree, tracking all partial programs with a stack; (Right) Execute all decoded programs, then returning the closest analog which minimizes distance to $\boldsymbol{x}$.

### 3.4 BI-LEVEL SYNTAX-SEMANTIC SYNTHESIZABLE MOLECULAR DESIGN

The task of synthesizable molecule design is to find a program $P$ and building blocks $B$ whose $M_*$ maximizes a property of interest. Given the learned policies from synthetic planning, we apply the inner loop procedure $F\colon \mathcal{X} \times \mathcal{T} \to \mathcal{P} \times \mathcal{B}^*$ as a surrogate, casting the optimization problem as one over the joint design space $\mathcal{X} \times \mathcal{T}$. We approach this problem with a genetic algorithm (GA) over fingerprint space $\mathcal{X}$ that leverages this extra dimension of syntax $\mathcal{T}$ (Figure 2 (Right)). The seed population is obtained by sampling random bit vectors. To generate a child $\boldsymbol{x}$ from two parents $\boldsymbol{x}_1$ and $\boldsymbol{x}_2$, we combine *semantic* crossover with *syntactical* mutation, reminiscent of our bi-level approach for analog generation:

**Semantic crossover:** We first generate $\boldsymbol{x}_*$ by combining bits from both $\boldsymbol{x}_1$ and $\boldsymbol{x}_2$ and possibly mutating a small number of bits of the result.

**Syntactic mutation:** We set $T = \tau(\boldsymbol{x}_*)$ and apply random edit(s) to obtain a perturbed tree $T'$. Together, $(\boldsymbol{x}_*, T)$ and $(\boldsymbol{x}_*, T')$ form a sibling pool. Applying the surrogate $F$ to each child gives two sibling SMILES that are turned into two sibling fingerprints. We fit a Gaussian process on past individuals and select the sibling with the highest expected improvement as the favoured child $\boldsymbol{x}$.

Intuitively, semantic crossover optimizes for chemical semantics by combining existing ones from the mating pool, while syntactic mutation explores syntactic analogs of individuals of interest. Each child is then given a fitness score under the property oracle, and the top scoring unique individuals are retained into the next generation (i.e., elitist selection). Further details and hyperparameters are given in App. G.

## 4 EXPERIMENTS

### 4.1 EXPERIMENT SETUP

#### 4.1.1 DATA GENERATION

We use 91 reaction templates from Hartenfeller et al. (2011); Button et al. (2019) representative of common synthetic reactions. They consist primarily of ring formation and fusing reactions but also peripheral modifications. Starting from 147,505 purchaseable compounds from Enamine Building Blocks, we follow the same steps as Gao et al. (2021) to generate 600,000 synthetic trees. Filtering by QED $> 0.5$ of the product molecules leaves 227,808 synthetic trees (136,684 for training, 45,563 for validation, and 45,561 for testing), which are then preprocessed into programs to construct our final datasets. We bootstrap our set of syntactic templates based on those observed in the training set, resulting in 1117 syntactic skeleton classes. Additional statistics on these syntactic templates and insights on their coverage are given in App. A. Further analyses of the structure-property relationship and a detailed case study are given in App. C.

### 4.1.2 BASELINES

We evaluate against all 25 molecular optimization methods benchmarked in the large-scale study by Gao et al. (2022). Most of these are divided into three categories based on the molecular representation used: string, graph, or synthesis trees. Synthesis methods restrict the design space to only products of robust template-based reactions, so for fair comparison, we also consider intra-category rankings. We report the synthetic accessibility (SA) score (Ertl & Schuffenhauer, 2009) of the optimized molecules to cross-verify the synthesizability of synthesis-based methods as well a investigate the performance trade-off imposed by constraining for template-compatible synthesizability.

### 4.1.3 COMPUTATIONAL EFFICIENCY

**Constructing** $\mathcal{D}$. Alg. 1 suggests $|\mathcal{D}| = O(2^{\max_\mathcal{T} |T|}|\mathcal{D}_0|)$ because we compute all the (closed under parent($\cdot$)) masks per $P \in \mathcal{D}_0$. However, we don't need to explicitly store $\mathcal{D}$. We only need to precompute the (closed under parent($\cdot$)) masks for each $T \in \hat{\mathcal{T}}_k$ ($k$ fixed and small), so the running time is $O(\sum_{T \in \hat{\mathcal{T}}_k} 2^{|T|})$, independent of $\mathcal{D}_0$. Besides, the actual number of masks is far smaller than $2^{|T|}$, and high $|T|$ is less represented in $\mathcal{D}_0$, since large programs are less likely to be sampled, so in practice $|\mathcal{D}|$ is much smaller than $2^{\max_\mathcal{T} |T|}|\mathcal{D}_0|$. Detailed statistics are given in App A.

**Training with** $\mathcal{D}$. During training, we flat-index into $\mathcal{D}_0$ and their precomputed masks to perform a pass over $\mathcal{D}$. Despite the larger dataset, we find the total training steps to actually be comparable with Gao et al. (2021). To scale to larger $k$, we propose a stratified sampling strategy that does only a $O(\mathcal{D}_0)$ pass per epoch, has positive support over $\mathcal{D}$, and represents each $(P, B) \in \mathcal{D}_0$ equally. The idea is to sample a constant number of masks $C$ per $P, B \in \mathcal{D}_0$, and re-sample each epoch. We show this has complexity linear to the *size* of $\mathcal{D}_0$ per epoch but empirically converges in much fewer training steps than SynNet for $C = 1$. Details are in App F.7. We further include an ablation study on the downstream task performance of this simplified training strategy in App F.5.

### 4.1.4 EVALUATION METRICS

**Synthesizable analog generation.** We evaluate the ability to generate a diverse set of structural analogs to a given input molecule using the Recovery Rate (RR, whether the most similar analog is exactly the target), Average Similarity (as measured by Tanimoto distance to the input), SA score, and Internal Diversity (average pairwise Tanimoto distance).

**Synthesizable molecule design.** We evaluate our method's ability to optimize 15 oracle functions (Huang et al., 2022) relevant to real-world drug discovery:

1. **Bioactivity predictors** (GSK3$\beta$, JNK3, DRD2) that estimate responses against targets related to the pathogenesis of various diseases such as Alzheimer's disease (Koch et al., 2015) based on experimental data (Sun et al., 2017), and whose inhibitors are the basis for many antipsychotics and have shown promise for treating diseases like Parkinson's schizophrenia and bipolar disorder (Madras, 2013).

2. **Structural profiles** (Median1, Median2, Rediscovery) that primarily focus on maximizing structural similarity to multiple molecules, which is useful for designing molecules fitting a more multifaceted structural profile (Brown et al., 2004). The rediscovery oracle focuses on hit expansion around a specific drug.

3. **Multi-property objectives** (Osimertinib, 6 others) that use real drugs as a basis for optimizing additional pharmacological properties, mimicking real-world drug discovery.

4. **Docking Simulations** ($M^{pro}$, DRD3) against $M^{pro}$, the main protease of SARS-Cov-2, and DRD3, which has its own leaderboard with a particular focus on sample efficiency.

In addition to the average score of the top $k$ molecules, we particularly focus on sample efficiency, i.e., the top-$k$ AUC as described in Gao et al. (2022).

### 4.2 RESULTS ON SYNTHESIZABLE ANALOG GENERATION

In Table 1, we see our method outperforming SynNet across both dimensions of similarity (how "analog" compounds are) and diversity (how different the compounds are). Additionally, our method

Table 1: We generate 5 unique analog molecules conditioned on an input molecule $M$ and sort them by decreasing similarity to $M$. For SynNet, we follow their beam search strategy and produce analogs using the top 5 beams. For Ours ($\tau$), we sample the top 5 syntactic templates from $\tau$. Ours ($\tau$, rev) is the same except we use a bottom-up decoding process, and it is included as an ablation for Section 4.4.1. Then, we evaluate how similar, diverse, and structurally simple the first $k$ molecules are. The best method is bolded. For three-way comparisons, the second best method is underlined.

| Dataset | Method | RR ↑ | Avg. Sim. ↑ | | | SA ↓ | | | Diversity ↑ | |
|---|---|---|---|---|---|---|---|---|---|---|
| | | | Top-1 | Top-3 | Top-5 | Top-1 | Top-3 | Top-5 | Top-3 | Top-5 |
| Train Set | Ours ($\tau$, rev) | 79.3 % | 0.923 | 0.632 | 0.569 | **3.072** | **2.795** | **2.716** | **0.615** | **0.657** |
| | Ours ($\tau$) | **88.1%** | **0.958** | **0.704** | **0.626** | 3.099 | 2.928 | 2.852 | 0.532 | 0.615 |
| Test Set | SynNet | 46.3% | 0.766 | **0.622** | **0.566** | 3.108 | 3.057 | 3.035 | 0.525 | 0.584 |
| | Ours ($\tau$, rev) | 40.8 % | 0.749 | 0.548 | 0.487 | **2.970** | **2.743** | **2.659** | **0.640** | **0.685** |
| | Ours ($\tau$) | **52.3%** | **0.799** | 0.588 | 0.548 | 3.075 | 2.895 | 2.856 | 0.609 | 0.653 |
| ChEMBL | SynNet | 4.9% | 0.499, | 0.436, | 0.394 | 2.669, | 2.685, | 2.697 | 0.644, | 0.693 |
| | Ours ($\tau$) | 7.6% | 0.531, | 0.443, | 0.396 | 2.544, | 2.510, | 2.460 | 0.675, | 0.727 |
| | Ours (MCMC) | **9.2%** | **0.532,** | **0.486,** | **0.432** | **2.364,** | **2.310,** | **2.263** | **0.765,** | **0.759** |

achieves lower SA Score, which is a proxy for synthetic accessibility that rewards simpler molecules. Guided by a set of simple yet expressive syntactic templates, our model simultaneously produces more diverse and structurally *simple* molecules without sacrificing one for the other. Additionally, our policy network is well-suited to navigate these simple yet horizon structures, enabling a 6% higher reconstruction accuracy after training on the same dataset. Combining these three dimensions, we can conclude our method is the superior one for the task of synthesizable analog generation. To better understand which design choices are responsible for the performance, we provided a comprehensive analysis of the policy network in App. F. We begin in App. F.3 by elaborating on the main distinction of our method vs existing works, highlight the novelty of our formulation, and motivate an auxiliary training task that takes inspiration from cutting-edge ideas in inductive program synthesis. We then perform several key ablations in App. F.4, using concrete examples to highlight success and failure cases. Lastly, we perform a step-by-step walkthrough of our decoding algorithm in App. F.8, visualizing the evolution of attention weights to showcase the full-horizon awareness of our surrogate and the dynamic incorporation of new information. These analyses shed insights into why our surrogate works, and points to future extensions to make it even better.

Table 2: Our model's average performance across 13 TDC oracles compared to baselines compiled in Gao et al. (2022). We limit to 1000 oracle calls each run and normalize oracle outputs to $[0, 1]$. We report to mean score, AUC, and SA scores (Ertl & Schuffenhauer, 2009) of the top 10 molecules. Methods are categorized by Gao et al. (2022) and, for brevity, we display only the top three algorithms within each category with respect their AUC. The best method per column is bolded and the best synthesis-based method is underlined. See App. H for the full results and experiment details.

| Category | Method | Score ↑ | | AUC ↑ | | SA ↓ | |
|---|---|---|---|---|---|---|---|
| | | Value | Rank | Value | Rank | Value | Rank |
| Screening | Screening | 0.426 | 20 | 0.377 | 20 | 3.097 | 8 |
| | MolPAL | 0.472 | 16 | 0.444 | 15 | 3.018 | 4 |
| String | REINVENT | 0.697 | 2 | 0.607 | 2 | 3.415 | 9 |
| | REINVENT-SELFIES | 0.682 | 3 | 0.578 | 4 | 3.791 | 15 |
| | STONED | 0.609 | 8 | 0.555 | 6 | 5.550 | 24 |
| | 7 rows omitted $\cdots$ | | | | | | |
| Graph | Graph-GA | **0.701** | 1 | 0.601 | 3 | 3.982 | 17 |
| | GPBO | 0.642 | 6 | 0.570 | 5 | 3.954 | 16 |
| | DST | 0.555 | 10 | 0.479 | 11 | 4.146 | 20 |
| | 7 rows omitted $\cdots$ | | | | | | |
| Synthesis | SynNet | 0.578 | 9 | 0.545 | 7 | 3.075 | 6 |
| | DoG-Gen | 0.634 | 7 | 0.511 | 9 | 2.793 | 2 |
| | DoG-AE | 0.460 | 18 | 0.450 | 14 | 2.857 | 3 |
| | Ours | 0.670 | 4 | **0.608** | 1 | **2.739** | 1 |

Table 3: AutoDock Vina scores against DRD3 and $M^{pro}$, limited to 5000 oracle calls. For ZINC (Screening), we use numbers from TDC's DRD3 Leaderboard, and for SynNet, we report both their paper's numbers (*) and our reproduced results. We also report the top 3 binders for $M^{pro}$ for the real-world case study in App. I.

| Target | Method | #Calls | Score ↑ | | | | | AUC ↑ | SA ↓ |
| | | | 1st | 2nd | 3rd | Top-10 | Top-100 | Top-10 | Top-10 |
|---|---|---|---|---|---|---|---|---|---|
| DRD3 | ZINC | — | 12.8 | — | — | 12.59 | 12.08 | — | — |
| | SynNet* | 5000 | 12.3 | — | — | 12.02 | 11.13 | — | — |
| | SynNet | 5000 | 10.8 | 10.4 | 10.3 | 10.30 | 9.20 | 9.55 | 2.59 |
| | Ours | 5000 | **13.7** | **13.1** | **13.1** | **13.01** | **12.13** | **11.91** | **2.13** |
| $M^{pro}$ | SynNet | 5000 | 8.3 | 8.3 | 8.2 | 8.09 | 7.46 | 7.60 | **2.27** |
| | Ours | 5000 | **9.9** | **9.7** | **9.7** | **9.54** | **9.02** | **9.01** | 2.59 |
| | SynNet* | — | 10.5 | 9.3 | 9.3 | — | — | — | — |
| | Ours | 10000 | **10.8** | **10.7** | **10.6** | — | — | — | — |

Table 4: (Left) Ablation of sibling pool generation strategies on JNK3: (edits) mutates the syntax, ($\tau$) uses the top skeletons predicted from $\tau$, and (flips) doesn't consider skeleton and instead flips random bits in the fingerprint. (Right) Ablation of SynNet with our Bayesian optimization (BO) acquisition over a sibling pool generated by beam search. **Seeds** and **All** are the average scores of the initial and final populations.

| | 1st | 2nd | 3rd | Top-10 | Top-100 | Diversity |
|---|---|---|---|---|---|---|
| Ours (edits) | **0.88** | **0.88** | **0.87** | **0.86** | **0.8** | **0.61** |
| Ours ($\tau$) | **0.88** | **0.88** | **0.87** | 0.83 | 0.74 | 0.55 |
| Ours (flips) | 0.87 | 0.87 | 0.86 | 0.84 | 0.77 | 0.49 |

| Oracle | Method | Top-1 | Top-10 | Top-100 | Seeds | All |
|---|---|---|---|---|---|---|
| GSK3$\beta$ | SynNet | 0.94 | 0.907 | 0.815 | $0.050 \pm 0.051$ | $0.803 \pm 0.041$ |
| | SynNet + BO | 0.85 | 0.684 | 0.471 | $0.013 \pm 0.024$ | $0.447 \pm 0.090$ |
| | Ours | **0.98** | **0.967** | **0.944** | $0.074 \pm 0.055$ | $\mathbf{0.941 \pm 0.012}$ |
| JNK3 | SynNet | 0.80 | 0.758 | 0.719 | $0.032 \pm 0.025$ | $0.715 \pm 0.017$ |
| | SynNet + BO | 0.31 | 0.241 | 0.143 | $0.006 \pm 0.012$ | $0.134 \pm 0.039$ |
| | Ours | **0.88** | **0.862** | **0.800** | $0.059 \pm 0.053$ | $\mathbf{0.792 \pm 0.030}$ |
| DRD2 | SynNet | **1.000** | **1.000** | 0.998 | $0.007 \pm 0.018$ | $0.996 \pm 0.003$ |
| | SynNet + BO | 0.982 | 0.963 | 0.722 | $0.005 \pm 0.018$ | $0.672 \pm 0.147$ |
| | Ours | **1.000** | **1.000** | **1.000** | $0.024 \pm 0.056$ | $\mathbf{1.000 \pm 0.000}$ |

## 4.3 RESULTS ON SYNTHESIZABLE MOLECULE DESIGN

In Table 2, we see our method outperforming *all* synthesis-based methods on average across the 13 TDC oracles for all considered metrics – average score, AUC, and SA score. Surprisingly, our method stays competitive with the state-of-the-art string and graph methods in terms of average score (ranking 4th) but being considerably more sample-efficient at finding the top molecules (ranking 1st for top-1/10 AUC). We see evidence that a synthesizability-constrained design space does not sacrifice end performance when reaping benefits of enhanced synthetic accessibility and sample efficiency.

The AutoDock Vina scores reflect our method's strength in real-world ligand design tasks. Our best binders against $M^{pro}$ in Table 3 are significantly better than nearly all known inhibitors from virtual screening or literature (Ghahremanpour et al., 2020; Zhang et al., 2021). For example, Zhang et al. (2021) report a best score of $-8.5$. Our best binders against DRD3 also rank us 3rd on the TDC Leaderboard (as of Aug. 2024). We present additional analysis of the best binders for our two docking targets in App. I.

## 4.4 ABLATION STUDIES

In Tables 1 and 4, we analyze findings from four carefully designed ablation studies (1 in 4.4.1, 2 & 3 in 4.4.2, 4 in 4.4.3) to justify the key design decisions that differentiate our method from the predecessor SynNet as well as other synthesis-based methods that have similar modules.

### 4.4.1 Top-down vs Bottom-up Decoding.

Should we decode top-down or bottom-up? This separation exists between retrosynthesis vs molecular design. Baselines that serialize the construction of synthesis trees (Bradshaw et al., 2019; 2020; Gao et al., 2021) adopt the latter. Given only knowledge of the target molecule, the model has to predict the first building block, the first reaction, and so on. Our empirical findings confirm observations by Gao et al. (2021) that the first few steps are difficult, due to inherent ambiguity – multiple valid choices exist. Our method resolves this bottleneck by reformulating the (Markov) state as partial

syntax trees, where holes are reactions and building blocks left to predict. This state captures the horizon structure, so we can learn tailored policies for the fixed horizon. We reintroduce the inductive bias of retrosynthetic analysis to procedural synthesis. Conditioned on the syntax (skeleton), we show a top-down filling order outperforms bottom-up, with two intuitive explanations: (1) there are orders of magnitude less reactions than building blocks (91 vs. 147,505) and (2) it is easier to reason backwards from the specification (target molecule) which reactions lead to the product. To demonstrate these factors compensate for any gains from bottom-up pruning of applicable reactions, we perform an additional ablation in Table 1. Instead of the MDP enforcing we fill in a skeleton top-down, we fill the skeleton from the bottom-up. We retrain the model by pretraining on inverted masks, and decode by following every possible topological order of the skeleton with edges reversed. The results show this cannot reconstruct the training data as well and struggles on generalization.

### 4.4.2 How Analog Generation Capabilities Translate to Design Capabilities.

Why does our superior analog generation capability translate to better performance when used as an offspring generator within molecule optimization? An ideal surrogate takes as input a fingerprint and outputs multiple synthesizable analogs, creating diverse offspring(s) that balance local neighborhood exploration with global exploitation. SynNet does the former by mutating the fingerprint directly, whereas the key insight of our syntax-guided method is to mutate the syntactic skeleton instead, doing so via editing mutations. Table 4 (Left) shows edit-based mutations are superior to the top recognition strategy used for analog generation ((Skeleton) in Table 1) and the trivial strategy of ignoring the skeleton and flipping individual bits to obtain siblings. This suggests edits to the skeleton better preserves the locality bias within the GA. Higher population diversity and average scores for $k \in \{10, 100\}$ suggest the same symbiotic relationship between diversity and similarity in analog generation is also the key enabler to better GA optimization. Our GA benefits from an inner loop sibling acquisition within the crossover operation, acquiring the highest expected improvement sibling to expend an oracle call on. It can be argued this extra mechanism is why our method gets better results and makes for an unfair comparison with SynNet, or that this is a method-agnostic hack to improve GA performance. In Table 4 (Right), we show SynNet endowed with a similar mechanism in its crossover operation (generate an offspring pool using the top beams then apply a BO acquisition step on top) didn't improve, but actually downgraded the performance. We hypothesize that SynNet's optimization trajectory is *derailed* by the additional variation to its sibling pool, reducing local movements within the output space that a syntactic editing approach naturally preserves. Thus, we believe the performance gains of this mechanism is *unlocked* by our syntax-driven approach.

### 4.4.3 Extrapolation to Unseen Templates.

How dependent is our framework on the initial set of syntactic templates? Can SynthesisNet generalize to more templates? Our ablation study in App. B evaluate whether the model can extrapolate to new template classes. We investigate: 1) how well SynthesisNet extrapolates to programs whose structural template was unseen during training, 2) how well the framework can incorporate new templates at test time, and 3) how robust overall task performance is to them. To answer these questions, we hold out $\approx 25\%$ of $\hat{\mathcal{T}}$ to be the test set, remove all programs belonging to those templates, retrain, and analyze changes in task performances, both holistic results and results specific to the held-out templates. Our results reveal minor performance drop, and in some instances, *improved* results along some metrics. Our analysis reveals the surrogate model is robust and unsensitive to missing templates, and the framework can also incorporate unseen templates in the online phases of the downstream tasks. We encourage future works to study the exact scaling laws between templates and performance, and we hope our initial findings unlock new directions for scaling up our solutions to achieve greater coverage and impact.

## 5 Discussion & Conclusion

We reconceptualize synthesis pathway design using a program synthesis approach, introducing a bi-level framework that separates the syntactical skeleton of a synthetic tree from its chemical semantics. Our learning algorithms leverage the tree horizon structure, improving performance on key metrics of analog generation and de novo design. By decoupling syntax and semantics, we effectively navigate a rich design space, integrating design and synthesis into a single workflow that reduces discovery cycle time. Our framework offers control over synthesis resources and biases towards simpler solutions, with the exciting prospect of integration with autonomous synthesis platforms (Coley et al., 2019).

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
