# A SYNTACTIC TEMPLATES

Syntactic templates form the essential ingredients for syntax-guided synthesis, as they significantly reduce the number of possible programs. In practice, syntactical templates are provided by users who operate with real-world constraints or experts who can help narrow the search space to *desirable* templates. The exact criteria for selecting templates are problem-dependent. To prove our concept in a more generalizable workflow, we bootstrap our set of syntactic templates $\hat{\mathcal{T}}$ in a data-driven way by obtaining the syntactic templates present in the training set. We then simulate real-world constraints by setting $\hat{\mathcal{T}}_k \leftarrow \{T \in \hat{\mathcal{T}} \mid T \text{ has at most } k \text{ internal nodes}\}$ and optimize within the induced design space $\partial \hat{\mathcal{P}}_k$ (Section 3.3.2). We tabulate summary statistics in Table 5 for the number of unique syntactic templates and the number of topological orders. We see that the empirical distribution is biased towards *simpler* syntactic templates, which reflects real-world constraints and is a key enabler of our amortized approach. We train the parameters $(\Theta, \Phi, \Omega)_k$ of our policies $(\tau, \pi_{\mathcal{R}}, \pi_{\mathcal{B}})$, respectively, for $k = 3, 4, 5, 6$ on our (pre)training dataset $\mathcal{D}$. For samples in $\mathcal{D}$ with more than $k > 6$ reactions, we snap it to the closest $T \in \hat{\mathcal{T}}_k$ according to the tree edit distance. We find $k = 3$ using full topological decoding (illustration in Figure 3) is best for Synthesizable Analog Generation and $k = 4$ with random sampling of the decoding beams is a good compromise between accuracy and efficiency for Synthesizable Molecule Design. We also note that the number of unique templates grows sub-exponentially, and in fact the number of templates for a fixed number of reactions starts diminishing for $k > 6$. To make sure this does not cause issues, we ensured there is still sufficient coverage to formulate a Markov Chain on $\hat{\mathcal{T}}_k$, which is crucial for our bilevel algorithms. For example, Figure 4 visualizes the empirical proposal distribution $J(T_1, T_2), \forall T_1, T_2 \in \hat{\mathcal{T}}_4 \times \hat{\mathcal{T}}_4$. Importantly, key hyperparameters like $\beta$ and $n_{\text{edits}}$ enable control over exploration vs exploitation.

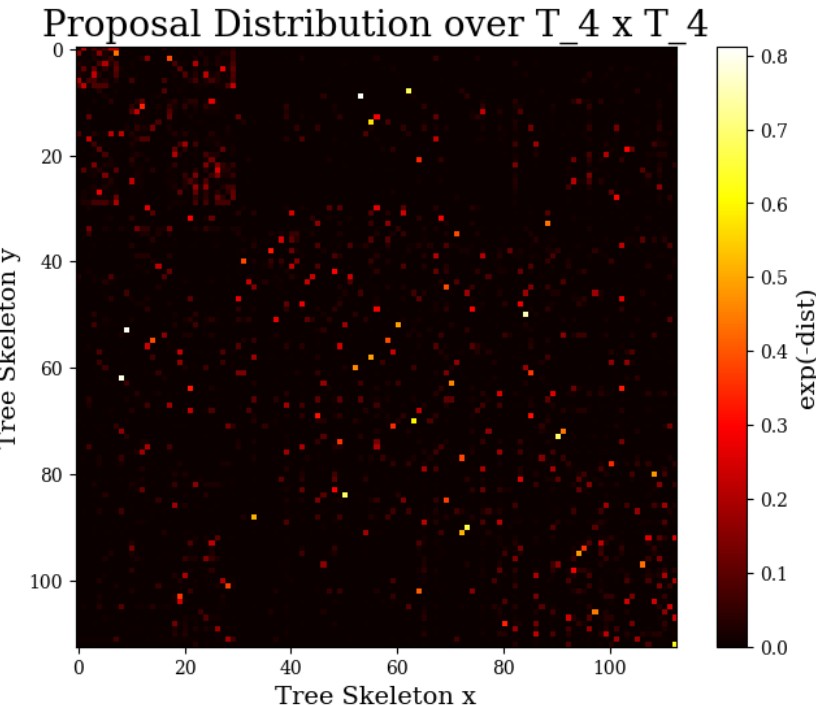

Figure 4: We adopt the tree edit distance as the dist function. We see that $\hat{\mathcal{T}}_4$ has sufficient transition coverage for bootstrapping our space of syntactic templates.

| # Rxns | $|\hat{\mathcal{T}}_{k\setminus k-1}|$ $(|\mathcal{T}_{k\setminus k-1}|)$ | # Topo. Orders (Max, Mean, Std)$_{\hat{\mathcal{T}}_{k\setminus k-1}}$ |
|---|---|---|
| 1 | 2 (2) | 2, 1.5, 0.5 |
| 2 | 6 (6) | 8, 4.17, 2.79 |
| 3 | 22 (22) | 80, 19.59, 20.55 |
| 4 | 83 (90) | 896, 152.02, 215.53 |
| 5 | 209 (394) | 19200, 2506.25, 3705.77 |

(a)

| # Rxns | $|\hat{\mathcal{T}}_{k\setminus k-1}|$ |
|---|---|
| 6 | 298 |
| 7 | 243 |
| 8 | 112 |
| 9 | 63 |
| 10 | 42 |
| 11 | 22 |
| 12 | 11 |
| 13 | 4 |
| 14 | 2 |

(b)

| # Rxns | # Topo. Masks ()$_{\hat{\mathcal{T}}_{k\setminus k-1}}$ | # Topo. Masks ()$_{\mathcal{T}_{k\setminus k-1}}$ |
|---|---|---|
| 1 | 5, 4, 1 | 5, 4, 1 |
| 2 | 11, 7.67, 2.56 | 11, 7.67, 2.56 |
| 3 | 26, 14.36, 5.86 | 26, 14.36, 5.86 |
| 4 | 56, 27.99, 12.47 | 56, 26.73, 12.78 |
| 5 | 131, 65.07, 26.36 | 131, 49.74, 27.09 |
| 6 | 287, 165.12, 61.43 | 287, 92.67, 56.29 |

(c)

Figure 5: (a) Summary statistics of the number of syntactic templates (both empirical and theoretically possible) and possible topological decoding node orders for $k = 1, 2, \ldots, 5$; (b) Summary statistics for only the number of syntactic templates since enumerating all topological sorts becomes intractable; (c) Summary statistics for the number of topological masks (subset of nodes closed under parent(.))

# B  EXTRAPOLATION TO UNSEEN TEMPLATES

## B.1  DATA PREPARATION

In this section, we investigate whether SynthesisNet can extrapolate to unseen templates, and effectively incorporate them for synthesizable analog generation and molecular design. We setup an ablation study as follows:

1. Reverse sort the templates by frequency using our dataset $\mathcal{D}_0$.

2. Collect every *fourth* template into a hold-out set $\mathcal{T}_{\text{test}}$ for $k = 4$.

3. Construct $\mathcal{D}_0' := \{(P, B) \in \mathcal{D}_0 \mid T_{P,B} \in \mathcal{T}_4 \setminus \mathcal{T}_{\text{test}}\}$ where $T_{P,B} \in \mathcal{T}$ is the syntactical template of $(P, B)$.

4. Run Algo. 1 on $\mathcal{D}_0'$ to obtain $\mathcal{D}'$.

5. Train ablation policy networks $\{\pi_{\mathcal{B}}', \pi_{\mathcal{R}}'\}$ using $\mathcal{D}'$.

6. Evaluate task performances with same $\tau$ as before.

We select hold-out templates in a frequency-stratified manner, ensuring the frequency distribution of $\hat{\mathcal{T}}_{\text{test}}$ is similar to that of $\hat{\mathcal{T}}$. Since smaller templates appear more frequently, the sizes of of templates are also indirectly stratified this way. Since we choose the least frequent from each consecutive group of 4 (Step 2), we note on average templates in $\hat{\mathcal{T}}_{\text{test}}$ tend to be slightly larger than $\hat{\mathcal{T}}$, so results for test templates may be lower in Table 5.

## B.2  RESULTS ON SYNTHESIZABLE ANALOG GENERATION

We evaluate the synthesizable analog task performances using ablation networks. We want to test whether the ablation networks integrate effectively with templates outside its structural support. Thus, we use the same $\tau$ as before for the $\tau$ experiments in Table 5, allowing access to the full template set but forcing $\{\pi_{\mathcal{B}}', \pi_{\mathcal{R}}'\}$ to extrapolate when performing inference over $\hat{\mathcal{T}}_{\text{test}}$.

Table 5: Apart from swapping out the policy networks, we use the same experimental setup as Table 1. For fair comparison, we also retrained Ours with $k = 4$ templates (whereas Table 1 used $k = 3$).

| Dataset | Method | RR ↑ | Avg. Sim. ↑ | | | SA ↓ | | | Diversity ↑ | |
|---------|--------|------|-------|-------|-------|-------|-------|-------|-------|-------|
| | | | Top-1 | Top-3 | Top-5 | Top-1 | Top-3 | Top-5 | Top-3 | Top-5 |
| Test Set | Ours:EP ($\tau$) | 52% | 0.815 | 0.616 | 0.548 | 3.140 | 2.964 | 2.892 | 0.585 | 0.646 |
| | Ours ($\tau$) | 56% | 0.827 | 0.633 | 0.555 | 3.100 | 3.019 | 2.918 | 0.543 | 0.628 |
| | Ours:EP $\mathcal{T}_{\text{test}}$ ($\tau$) | 20% | 0.636 | 0.539 | 0.473 | 2.844 | 3.023 | 2.987 | 0.564 | 0.675 |
| | Ours $\mathcal{T}_{\text{test}}$ ($\tau$) | 20% | 0.626 | 0.552 | 0.493 | 3.360 | 3.135 | 3.070 | 0.542 | 0.634 |
| ChEMBL | Ours ($\tau$) | 7.6% | 0.531 | 0.443 | 0.396 | 2.544 | 2.510 | 2.460 | 0.675 | 0.727 |
| | Ours (MCMC) | 9.2% | 0.532 | 0.486 | 0.432 | 2.364 | 2.310 | 2.263 | 0.765 | 0.759 |
| | Ours:EP (MCMC) | 8.5% | 0.519 | 0.421 | 0.367 | 2.644 | 2.420 | 2.382 | 0.618 | 0.640 |

The performance takes only a minor dip on the Test Set, compensating for slightly lower Avg. Sim. with higher Diversity and comparable SA. We further zoom in on the subset of $\mathcal{D}_0$ with structure among the held-out template classes. We emphasize this is very difficult for $\{\pi'_\mathcal{B}, \pi'_\mathcal{R}\}$ to do, which has not seen any examples from those structural classes. It actually appears Ours:EP has the slight edge over Ours on the held-out template set, with slightly better SA and greater diversity. We attribute this to a regularization effect induced by removing (slightly, due to Step 2.) more complex program structures. There are still enough complex templates left that this does not harm performance, highlighting the robustness of our model in this setting. We believe the fact the ablation model can maintain comparable performance implies the following:

- **Structural Extrapolation**: It is capable of inference of programs outside the structural support of its training distribution in this case.

- **Template Set Robustness**: Our model is not very sensitive to the default size of the template set, since using only 75% of it already brings it close to diminishing returns.

The dip in performance is more noticeable for the predominantly unsynthesizable dataset ChEMBL, with lower metrics across the board. We suspect the reason is due to ChEMBL containing more complex molecules that require longer synthetic routes. This is also apparent from the lower analog diversity, suggesting training on more template variety helps.

We believe the difficulty of the task (ratio of synthesizable vs unsynthesizable molecules) can inform whether the method is sensitive to the template set it sees during training. Similar ablation studies to this one can highlight when additional resources should be allocated to expanding the training set and when it is sufficient to simply incorporate more templates at test time.

## B.3 RESULTS ON SYNTHESIZABLE MOLECULAR DESIGN

Figure 6: We select the first Oracle from each Table in App. H to compare Ours with Ours (EP). Aside from the ablation networks, we use the same experimental settings as Table 2.

(a)

| | | | | GSK3$\beta$ | | | | | | | | | | Median 1 | | | | | | |
|---|---|---|---|---|---|---|---|---|---|---|---|---|---|---|---|---|---|---|---|---|
| | | | | Top 1 | | | Top 10 | | | Top 100 | | | | Top 1 | | | Top 10 | | | Top 100 |
| | category | Oracle Calls | Score | SA | AUC | Score | SA | AUC | Score | SA | AUC | Oracle Calls | Score | SA | AUC | Score | SA | AUC | Score | SA | AUC |
| Ours (EP) | synthesis | 6921 | **0.99** | **1.975** | 0.872 | 0.965 | 2.321 | 0.818 | 0.94 | **2.237** | 0.744 | 9050 | **0.4** | 4.12 | 0.358 | **0.344** | 4.434 | **0.305** | **0.301** | 4.394 | **0.257** |
| Ours | synthesis | 4886 | 0.98 | 2.045 | **0.891** | **0.967** | **2.302** | **0.848** | **0.944** | 2.27 | **0.778** | 8303 | **0.4** | **3.353** | **0.371** | 0.342 | **4.161** | **0.305** | 0.298 | **4.256** | 0.252 |

(b)

| | | | | Osimertinib MPO | | | | | | | | | | Perindopril MPO | | | | | | |
|---|---|---|---|---|---|---|---|---|---|---|---|---|---|---|---|---|---|---|---|---|
| | | | | Top 1 | | | Top 10 | | | Top 100 | | | | Top 1 | | | Top 10 | | | Top 100 |
| | category | Oracle Calls | Score | SA | AUC | Score | SA | AUC | Score | SA | AUC | Oracle Calls | Score | SA | AUC | Score | SA | AUC | Score | SA | AUC |
| Ours (EP) | synthesis | 10000 | 0.852 | 2.322 | **0.831** | **0.849** | 2.475 | **0.823** | **0.839** | 2.484 | **0.802** | 10000 | **0.626** | 3.382 | **0.562** | **0.598** | 3.375 | **0.54** | 0.56 | 3.349 | **0.509** |
| Ours | synthesis | 10000 | **0.859** | **2.263** | 0.826 | 0.847 | **2.21** | 0.81 | 0.832 | **2.249** | 0.769 | 10000 | 0.622 | 3.338 | 0.547 | 0.591 | 3.378 | 0.524 | **0.558** | **3.137** | 0.485 |

We also evaluate the synthesizable molecular design performances using ablation networks. We want to evaluate whether the ablation models can guide the optimization trajectory as a surrogate generator of synthesizable analogs. We see comparable performances across all four Oracles in Table 14, and for each Ours (EP) having the edge on some metrics while Ours having the edge on other metrics. This suggests both are capable enough to serve as the inner subroutine of our bilevel

genetic framework, although the models may have different biases on the kind of analogs it generates which affects the optimization trajectories. Since Ours (EP) may generate less structurally diverse analogs, it can converge in fewer Oracle calls, resulting in less Oracle calls and slightly higher AUCs. Meanwhile, Ours produce more diverse analogs, which enables the acquisition of higher confidence analogs. We see Ours to have the edge on SA across Median 1 (Top 1) and the MPO (Top 100) Oracles. This may be because higher confidence regions tend to be where the simpler molecules are, resulting in simpler analogs hence lower SA. Overall, the results show the robustness of our model.

## C  SYNTAX TREE RECOGNITION

In this section, we answer key questions like: (1) How does the relationship change with the addition of $\mathcal{T}$? (2) How strong is the correlation between $\mathcal{X}$ and $\mathcal{T}$? (3) How justified are the most confident predictions made by $\tau$? We investigate the relationship between $\mathcal{M}$ and $\mathcal{T}$. We seek to understand the extent to which the true mapping $\mathcal{M} \to \mathcal{T}$ is well-defined. The first part is quantitative analysis, and the second part is a qualitative study.

### C.1  T-SNE AND MDS PLOTS

We use the t-distributed stochastic neighbor embedding (t-SNE) on the final layer hidden representations of our MLP $\tau$ to visualize how our recognition model discriminates between molecules of different syntax tree classes. From Figure 7, we see the MLP is able to discriminate amongst the top 3 or 4 most popular skeleton classes, visually partitioning the representation space. However, beyond that the representations on the validation set begin to coalesce, i.e., the model begins overfitting.

Figure 7: t-SNE on molecules in top (3,4,5,6) skeleton classes

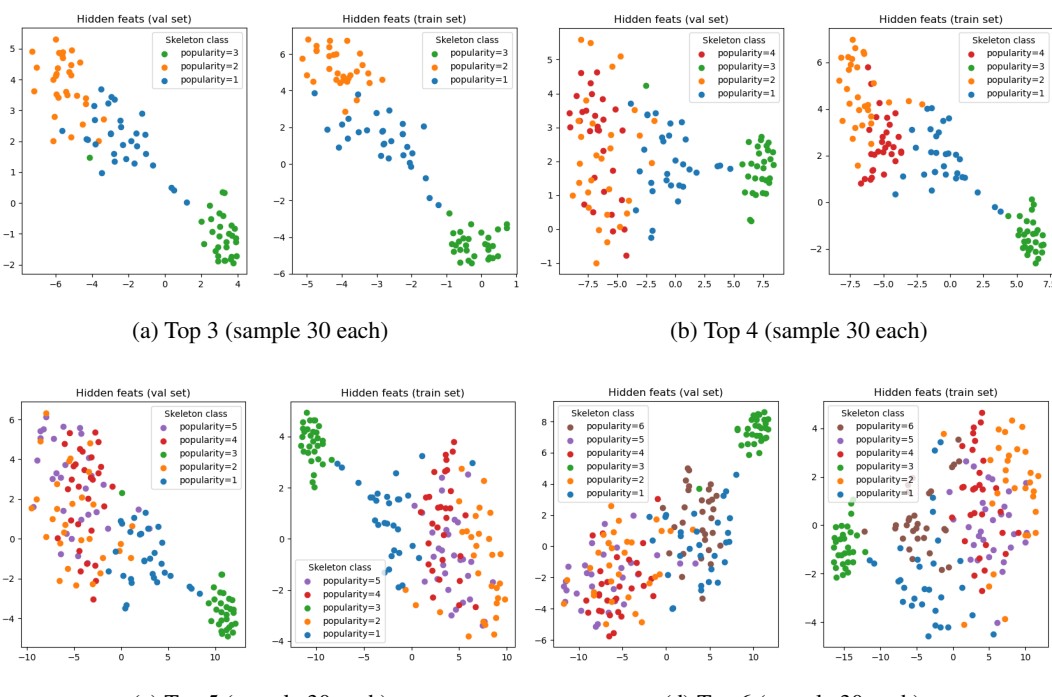

(a) Top 3 (sample 30 each)        (b) Top 4 (sample 30 each)

(c) Top 5 (sample 30 each)        (d) Top 6 (sample 30 each)

Since gradient descent is stochastic, we also use multi-dimensional scaling (MDS) using the Morgan Fingerprint Manhattan distance on a subset of our dataset to visualize the relative positioning between molecules of different syntax tree classes (sorted based on popularity). From the plots in Figure 9, we observe some interesting trends:

- **Similarly positioned points tend to have similar colors.**

Figure 8: MDS on molecules in top (10,20,100) skeleton classes

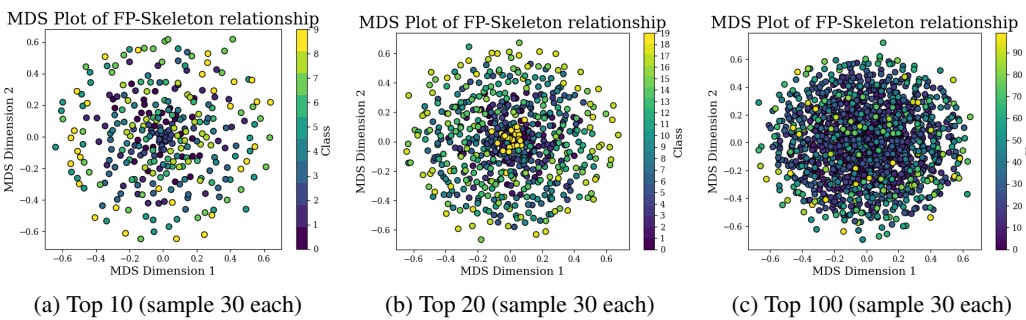

(a) Top 10 (sample 30 each)     (b) Top 20 (sample 30 each)     (c) Top 100 (sample 30 each)

- The darker end of the spectrum corresponding to the most popular classes generally cluster together in the middle.
- The classes do not form disjoint partitions in space. As the ranked popularity increases, the points tend to disperse outwards. There are exception classes, e.g., the yellow set of points in Figure 8b that cluster in the center.

Based on these findings, it's reasonable to conclude a recognition classifier by itself is overly naive. However, the useful inductive bias that similar molecules are more likely to share the same syntactic template indicates the **localness** property still holds. Our method is designed with this property in mind: we encourage iterative refinement of the syntactic template when doing analog generation.

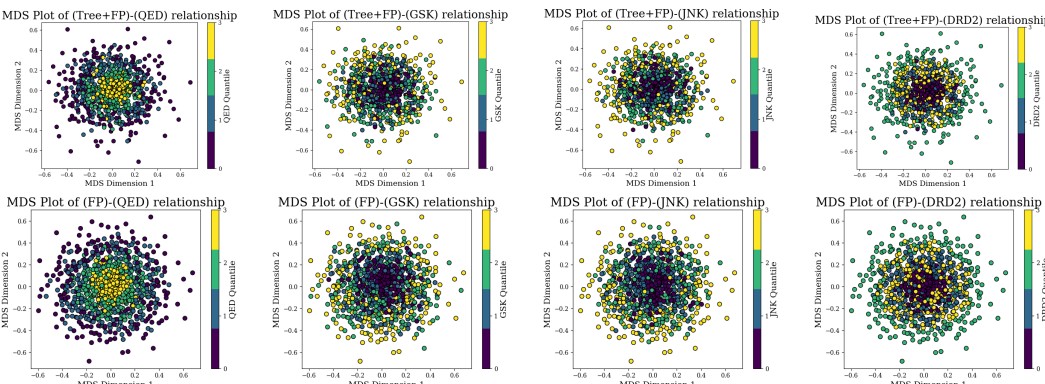

Figure 9: We visualize the structure-property relationship as a scatterplot of 2D structures vs property values. (Top) Structure is $\mathcal{X} \times \mathcal{T}$. We use MDS with the dissimilarity $d_{\mathcal{X} \times \mathcal{T}}((\boldsymbol{x}_1, T_1), (\boldsymbol{x}_2, T_2)) = ||\boldsymbol{x}_1 - \boldsymbol{x}_2||_1 + \text{Tree-Edit-Distance}(T_1, T_2)$. (Bottom) Structure is only $\mathcal{X}$.

We also use MDS to investigate the structure-property relationship to understand the joint effect $\mathcal{T}$ and $\mathcal{X}$ has on different properties of interest. As shown in Figure 9, we see overall, the functional landscape varies significantly from property to property, but the general trend is that decoupling $\mathcal{T}$ from $\mathcal{X}$ does not change the structure-property relationship much. Whereas analog generation requires a more granular understanding of the synergy between $\mathcal{X}$ and $\mathcal{T}$, molecular optimization does not. Instead, the evolutionary strategy should be kept fairly consistent between the original design space ($\mathcal{X}$) and ($\mathcal{X} \times \mathcal{T}$). However, the top row exhibits lower entropy, with the empirical distribution looking "less Gaussian". To capture this nuance, the evolutionary algorithm should combine both global and local optimization steps. We meet this observation with a bilevel optimization strategy that combines semantic crossover with syntactic mutation.

## C.2 EXPERT CASE STUDY

In this section, we enter the perspective of the recognition model learning the mapping from molecules to syntax tree skeletons. The core difference between this exercise and a common organic

chemistry exam question (Flynn, 2014) is the option to abstract out the specific chemistry. Since the syntax only determines the skeletal nature of the molecule, the specific low-level dynamics don't matter. As long as the model can pick up on skeletal similarities between molecules, it will be confident in its prediction. We did the following exercise to understand if cases where the recognition model is most confident on unseen molecules can be attributed to training examples. We took the following steps:

For each true skeleton class $T$

1. Inference the recognition model on 10 random validation set molecules belonging to $T$.
2. Pick the top 2 molecules the model was most confident belongs to $T$.
3. For each molecule $M$.
   1. Find the 2 nearest neighbors to $M$ belonging to $T$ in the training set.

Shown in Figures Figure 10 and Figure 11 is the output of these steps for a common skeleton class which requires two reaction steps.

Figure 10: COc1ncnc(N2C(=O)c3cc([N+](=O)[O-])c(O)cc3N=C2C2NC(=O)OC23CCC3)c1C which recognition model predicts is in its true class with 87.5% probability

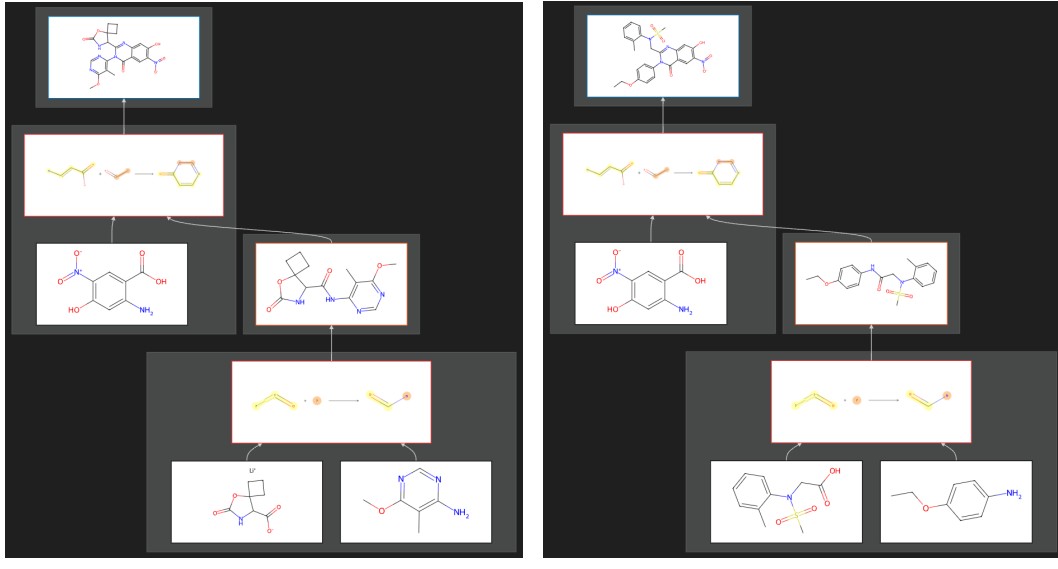

(a) Query molecule        (b) Nearest neighbor in training set

In Figure 10, we see that the query molecule's nearest neighbor is an output from the *same program* but different building blocks. Both feature the same core fused ring system involving a nitrogen. Given that the model has seen Figure 10b (and other similar instances), it should associate this core feature with a ring formation reaction step. Taking a step deeper, the respective precursors also share the commonality of having an amide linkage in the middle. Amides are key structural elements that the recognition model can identify. Both precursors underwent the same amide linkage formation step, despite the building blocks being different. Thus, the model's high confidence on the query molecule can be attributed directly to Figure 10b.

In Figure 11, there is more "depth" to the matter. We see a skeletal similarity across all three molecules: a nitrogen in the center with three substituents. Although it's noteworthy that the nitrogen participates in a sulfonamide group in all three cases, using this fact to inform the syntax tree would be a mistake. This is because in Figure 11a and Figure 11b, the sulfonamide group is the result of an explicit sulfonamide formation reaction, where a sulfonyl chloride reacts with an amine. However, in Figure 11c the sulfonamide group is already present in a building block. Thus, we see where the recognition model taking as input the circular fingerprint of this molecule could overfit. Nonetheless, the nitrogen with three substituents necessitates at least one reaction is required. The necessity for a

Figure 11: COC(Cc1ccccc1)CN(C1CCCOc2ccccc21)S(=O)(=O)Cc1ccon1 which recognition model predicts is in its true class with 86.2% probability

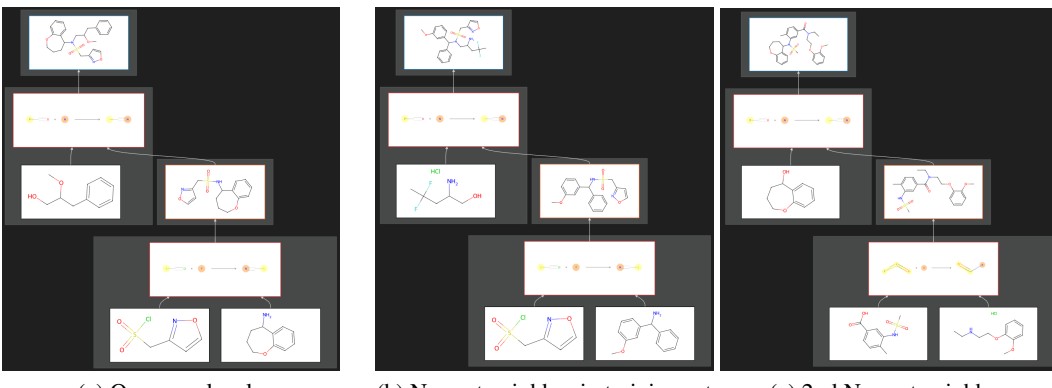

(a) Query molecule      (b) Nearest neighbor in training set      (c) 2nd Nearest neighbor

second reaction can be attributed to the ether linkage present in both Figure 11a and Figure 11c. The recognizer would be able to justify an additional reaction after it has seen the bicyclic ring structure joined with the sulfonamide group sufficiently many times before. In summary, the model will often be presented multiple complex motifs, but only a subset of them may be responsible for reaction steps. The exact number of reactions needed can only be determined via actually doing the search, but high-level indicators (such as the nitrogen with three substituents) allow the recognition model to abstract out the semantic details and construe a "first guess" of what the syntax tree is.

## D  EXPANDED RELATED WORKS

### D.1  BACKGROUND ON PROGRAM SYNTHESIS

Program synthesis is the problem of synthesizing a function $f$ from a set of primitives and operators to meet some correctness specification. For example, if we want to synthesize a program to find the max of two numbers, the correctness specification $\phi_{\max} \coloneqq f(x,y) \geq x \wedge f(x,y) \geq y \wedge (f(x,y) = x \vee f(x,y) = y)$. As our approach is inspired from ideas in program synthesis, we briefly cover some basic background. A program synthesis problem entails three things:

1. Background theory $\mathsf{T}$ which is the vocabulary for constructing formulas, a typical example being linear integer arithmetic: which has boolean and integer variables, boolean and integer constants, connectives ($\wedge, \vee, \neg, \rightarrow$), operators ($+$), comparisons ($\leq$), conditional (If-Then-Else)
2. Correctness specification: a logical formula involving the output of $f$ and $\mathsf{T}$
3. Set of expressions $L$ that $f$ can take on described by a context-free grammar $G_L$.

Program synthesis is often formulated as deducing a constructive proof to the statement: for all inputs, there exists an output such that $\phi$ holds. The constructive proof itself is then the program. At the low-level, program synthesis methods repeatedly calls a SAT solver with the logical formula $\neg \phi$. If UNSAT is returned, this means $f$ is valid. Syntax-guided synthesis (Alur et al., 2013; Schkufza et al., 2013) (SyGuS) is a framework for meeting the correctness specification with a syntactic template. Syntactic templates explicitly constrains $G_L$, significantly reducing the number of implementations $f$ can take on. Sketching is an example application where programmers can sketch the skeletal outline of a program for synthesizers to fill in the rest (Solar-Lezama et al., 2005). More directly related to our problem's formulation is inductive synthesis, which seeks to generate $f$ to match input/output examples. The problem of synthesis planning for a molecule $M$ is a special case of the programming-by-example paradigm, where we seek to synthesize a program consistent with a single input/output pair: $(\{B\}, M)$. Inductive synthesis search algorithms have been developed to search through the combinatorial space of derivations of $G_L$. In particular, stochastic inductive synthesis use techniques like MCMC to tackle complex synthesis problems where enumerative methods do not scale to. MCMC has been used to optimize for the opcodes in a program (Schkufza et al., 2013) or for the abstract syntax tree directly (Alur et al., 2013). In our case, the space of possible program

semantics is so large that we decouple the syntax from the semantics, performing stochastic synthesis over only the syntax trees. We also borrow ideas from functional program synthesis, where top-down strategies are preferred over bottom-up ones to better leverage the connection between a high-level specification and a concrete implementation (Polikarpova et al., 2016). Similar to how top-down synthesis enables aggressive pruning of the search space via type checking, retrosynthesis algorithms leverages the target molecule $M$ to prune the search space via template compatability checks.

## D.2 EXECUTION-GUIDED PROGRAM SYNTHESIS

We would like to note the distinction between our program synthesis formulation and other formulations. Retrosynthesis is essentially already guided by the execution state at every step. Each expansion in the search tree executes a deterministic reaction template to obtain the new intermediate molecule. Planners based on single-step models (Chen et al., 2020), for example, assume the Markov Property by training models to directly predict a template given *only* the intermediate (Torren-Peraire et al., 2024; Tu et al., 2022). In program synthesis, meanwhile, the state space is a set of partial programs with actions corresponding to growing the program. The execution of the program (or verification against the specification) does not happen until a complete program is obtained. In recent years, neural program synthesis methods found using auxiliary information in the form of the *execution state* of a program can help indirectly inform the search (Bunel et al., 2018; Chen et al., 2018; Ellis et al., 2019) since it gives a sense on what the program can compute so far. This insight does not apply to retrosynthesis, since retrosynthesis already executes on the fly. It also does not apply for the methods introduced in Section 2.2 that construct a synthetic tree in a bottom-up manner, for the same reason (the only difference is they use forward reaction templates, with a much smaller set of robust reaction templates) to obtain the execution state each step. However, as described in Section 3.3.1, our approach combines the computational advantages of restricting to a small set of forward reaction templates with the inductive bias of retrosynthetic analysis. Our policy is to predict *forward* reaction templates in a *top-down* manner. This formulation is common in top-down program synthesis, where an action corresponds to selecting a hole in the program. Similarly, our execution of the program does not happen until the tree is filled in. However, we leverage the insight that the execution state helps in an innovative way, as discussed in F.3.

## D.3 INSPIRATIONS FROM RETROSYNTHESIS AND ALTERNATE FORMULATIONS

We begin by elaborating the distinctions between retrosynthesis methods and methods for synthesizable molecular design. Then, we identify a few recent works from retrosynthesis that can inspire cross-pollination of ideas. Finally, we end with alternate formulations of the problem that are also valuable to consider for future cross-examination.

**Intended Use Case**

Retrosynthesis aims to find a synthetic route for a given target molecule, without reference to how the target molecule is obtained or further optimizations on the target molecule. The target molecule is a compound that may serve any application or use case that we will not get into here, but importantly it *is* the problem to solve. We refer readers to Gao & Coley (2020) for further descriptions.

Synthesizable molecular design aims to be a standalone molecular optimization workflow that explicitly constrains the design space to be synthetically accessible building blocks and reactions Vinkers et al. (2003). This is often coupled with property oracles that evaluate the designs, which guides the optimization towards parts of the design space with higher fitness Gao et al. (2022).

**MDP Formulation**

Retrosynthesis can be formulated as a tree-shaped MDP, where each state is a molecule (initial state being the target, terminal states being building blocks) and each action is a reversed ("retro") reaction. The tree shape of the MDP is due to the fact the retro reaction (action) produces a set of reactants (states) Liu et al. (2023). Retrosynthetic planners often tackle the MDP by combining a single-step model (predicting retro reactions) and a multi-step planner (e.g. A* search Liu et al. (2018), MCTS Segler & Waller (2017), depth-first search Kishimoto et al. (2019)). A solution to the MDP is a *tree* of actions, i.e. a synthetic tree, where all sequences of actions in this tree lead to terminal states.

Synthesizable molecular design feature a broader set of methods but can be defined as a discrete optimization problem over synthetic trees directly: $\arg\max_x f(F(x))$ where $x$ is a synthetic tree, $F$ the root molecule of $x$ and $f$ is the fitness function. As the design space is intractably large, prior approaches discussed in 2.3 formulate the problem as a serial MDP, where each state is a synthetic tree and each action an edit operation (add, merge, etc.) to the synthetic tree. Though simple, we argue such a formulation is ill-advised, for reasons we discussed and demonstrated in the main text. We use an alternate formulation, inspired by program synthesis, that considers each state as a partially sketched program and each action as completing a hole of the program.

**Learning Goal**

Retrosynthesis methods aim to learn a policy $\pi(a|s)$, where $s$ is a molecule and $a$ a retro reaction, which can be template-based or template-free (we won't go into that here). Traditional works learn the policy from public datasets of synthetic routes (e.g. USPTO), but recent works have explored novel strategies for learning $\pi$ by combining offline and online training. The offline training is usually done on a reaction dataset to initialize a policy network and/or reaction model. The online training iteratively adapts the policy network by acquiring more data using a planning algorithm, possibly guided by the current policy network. More specifically, Guo et al. (2024) uses MCTS to acquire data, inferring policy and value targets based on the node visit counts. Kim et al. (2021) uses a self-improvement strategy, reminiscent of AlphaGo Zero Silver et al. (2017), that trains independent reference forward and backward reaction models to control the quality and diversity of new reaction pathways acquired by the planner. Liu et al. (2023) follows a similar strategy, but decouples the synthesizability and cost of the value function. They also architect a two-branch policy network that uses a trainable single-step network to optimize the probabilities over reactions from the frozen reference network to better model real-world synthesis considerations. Like Guo et al. (2024), they use MCTS guided by the current policy network to acquire more data, creating a synergistic feedback loop that results in a holistic, trained policy network which can be plugged into multi-step planners.

Synthesizable molecular design methods, meanwhile, are relatively more spread out, with research going into problem formulation and algorithmic frameworks for tackling this more open-ended problem. In SynthesisNet, the policy learning is done entirely offline (as described in Sections 3.3.1 and 4.1.1) to amortize for the cost of searching during the actual online phases (MCMC and GA), but above techniques from retrosynthesis can also facilitate self-improving the surrogate network. One potential idea for generating more experiences is to take partial program examples from the existing data, use guided planning to complete those examples, then retrain the surrogate network with the augmented dataset. We leave the details for exciting future extensions of our work.

**Alternate Formulations**

Towards synthesizable molecular design, alternate formulations from recent developments can also be considered. Decision Transformer Chen et al. (2021) is a recent work that re-imagines offline RL as a conditional sequence modeling task. Notably, the model conditions on reward-to-go and the history for generating the next action. The Transformer architecture enables long-term modeling of the environment's dynamics, enabling credit assignment and relational modeling of its history. The offline setting tackled by Decision Transformer naturally aligns with our method, where we do (self-)supervised policy training from programs generated offline. Although the program structure carries important hierarchical information about the relations of building blocks and reactions to one another, it's worth considering whether the synthesis tree construction serialization protocol used by prior works Bradshaw et al. (2020); Gao et al. (2021); Luo et al. (2024); Gao et al. (2024) can be used to formulate a conditional sequence modeling problem. GFlowNet Bengio et al. (2021; 2023) is also a recent work that formulates a flow network over a tree-based MDP, where the incoming and outgoing flow are proportional the probability of subsequent actions and learned using flow matching objectives. The goal of this model is to learn amortized samplers for reward functions, producing both diverse and high-quality samples constructed in a step-by-step manner following an MDP environment. The learning occurs on offline trajectories with observed rewards. The formulation using reward functions can be a direction of future work for our framework, which currently only considers rewards in the online phases. However, if we had considered rewards to be provided upfront along with the data generation procedure, we can adopt GFlowNet to amortize the expensive work done by MCMC, and directly sample programs. We leave this study for future works. We believe cross-examining alternate formulations and recent methodologies to be essential for finding future inspirations for extending the innovation horizon of methods used to tackle synthesizable molecular design.

# E  DERIVATION OF GRAMMAR

We now define the grammar $G_{\mathcal{P}}$ describing the set of implementations our program can take on. A context-free grammar is a tuple $G_{\mathcal{P}} \coloneqq (\mathcal{N}, \Sigma, \mathcal{P}, \mathcal{X})$ that contains a set $\mathcal{N}$ of non-terminal symbols, a set $\Sigma$ of terminal symbols, a starting node $\mathcal{X}$, and a set of production rules which define how to expand non-terminal symbols. Recall we are given a set of reaction templates $\mathcal{R}$ and building blocks $\mathcal{B}$. Templates are either uni-molecular ($\coloneqq \mathcal{R}_1$) or bi-molecular ($\coloneqq \mathcal{R}_2$), such that $\mathcal{R} = \mathcal{R}_1 \cup \mathcal{R}_2$. In the original grammar, these take on the following:

1. **Starting symbol**: $T$

2. **Non-terminal symbols**: $R_1$, $R_2$, $B$

3. **Terminal symbols**:
    - $\{R \in \mathcal{R}_1\}$: Uni-molecular templates
    - $\{R \in \mathcal{R}_2\}$: Bi-molecular templates
    - $\{BB \in \mathcal{B}\}$: Building blocks

4. **Production rules**:
    1. $T \to R_1$
    2. $T \to R_2$
    3. $R_1 \to R(B)$ ($\forall R \in \mathcal{R}_1$)
    4. $R_1 \to R(R_1)$ ($\forall R \in \mathcal{R}_1$)
    5. $R_1 \to R(R_2)$ ($\forall R \in \mathcal{R}_1$)
    6. $\forall (X_1, X_2) \in \{\text{``}R_1\text{''}, \text{``}R_2\text{''}, \text{``}B\text{''}\} \times \{\text{``}R_1\text{''}, \text{``}R_2\text{''}, \text{``}B\text{''}\}$
        - $R_2 \to R(X_1, X_2)$ ($\forall R \in \mathcal{R}_2$)
    7. $B \to BB$ ($\forall BB \in \mathcal{B}$)

Example expressions derived from this grammar are "R3(R3(B1,B2),R2(B3))" and "R4(R1(B2,B1))" for the programs in Figure 1.

Identifying a retrosynthetic pathway can be formulated as the problem of searching through the derivations of this grammar conditioned on a target molecule. This unconstrained approach is extremely costly, since the number of possible derivations can explode.

In our syntax-guided grammar, we are interested in a finite set of syntax trees. The syntax tree of a program depicts how the resulting expression is derived by the grammar. These are either provided by an expert who has to meet experimental constraints, or specified via heuristics (e.g., maximum of $x$ reactions, limiting the tree depth to $y$). For example, the syntax-guided grammar for the set of trees with at most 2 reactions is specified as follows:

1. **Starting symbol**: $T$

2. **Non-terminal symbols**: $R_1$, $R_2$, $B$

3. **Terminal symbols**:
    - $\{R \in \mathcal{R}_1\}$: Uni-molecular templates
    - $\{R \in \mathcal{R}_2\}$: Bi-molecular templates
    - $\{BB \in \mathcal{B}\}$: Building blocks

4. **Production rules**:
    1. $T \to R_2(B, B)$
    2. $T \to R_1(B)$
    3. $T \to R_1(R_2(B, B))$
    4. $T \to R_1(R_1(B))$
    5. $T \to R_2(B, R_1(B))$
    6. $T \to R_2(B, R_2(B, B))$
    7. $T \to R_2(R_1(B), B)$
    8. $T \to R_2(R_2(B, B), B)$

9. $R_1 \to R$ $(\forall R \in \mathcal{R}_1)$
10. $R_2 \to R$ $(\forall R \in \mathcal{R}_2)$
11. $B \to BB$ $(\forall BB \in \mathcal{B})$

This significantly reduces the number of possible derivations, but two challenges remain:

- How can when pick the initial production rule when the number of syntax trees grow large? *We use an iterative refinement strategy, governed by a Markov Chain Process over the space of syntax trees. The simulation is initialized at the structure predicted from our recognition model Appendix C.*

- How can we use the inductive bias of retrosynthetic analysis when applying rules 9, 10, 11? *We formulate a finite horizon MDP over the space of partial programs, where the actions are restricted to decoding only frontier nodes. This topological order to decoding is consistent with the top-down problem solving done in retrosynthetic analysis. Furthermore, our pretraining and decoding algorithm enumerates all sequences consistent with topological order.*

These two questions are addressed by the design choices in Section 3.3.

## F  POLICY NETWORK

### F.1  FEATURIZATION

Our dataset $\mathcal{D}$ comprises partial programs $T \in \partial \mathcal{P}$ producing molecules $M$. Then, we compute node features $\boldsymbol{H}$ and labels $\boldsymbol{Y}$ as:

$$\boldsymbol{h}_n := [\text{FP}_{2048}(M), \text{BB}_{2048}(n), \text{RXN}(n)], \quad \boldsymbol{y}_n := \begin{cases} \text{RXN}(n), & \text{if } i \text{ is a reaction node,} \\ \text{BB}_{256}(n), & \text{otherwise,} \end{cases}$$

where $\text{FP}_d(\cdot)$ computes the $d$-bit radius 2 Morgan fingerprints, $\text{BB}_d(n) = \text{FP}_d(n_{\text{SMILES}})$ if $n$ is attributed with a building block from $\mathcal{B}$ or $\boldsymbol{0}_d$ otherwise and $\text{RXN}(n) = \text{one\_hot}_{91}(n_{\text{RXN\_ID}})$ if $n$ is attributed with a reaction from $\mathcal{R}$ or $\boldsymbol{0}_{91}$ otherwise.

If $\mathcal{N}(T)$ and $\mathcal{E}(T)$ denote the node and edge set of $T \in \partial\mathcal{P}$, then we define, for convenience:

$$\text{RXN}(T) := \{r \in \mathcal{N}(T) \mid \exists c, p \in \mathcal{N}(T) \text{ s.t. } (r, c) \in \mathcal{E}(T) \cap (p, r) \in \mathcal{E}(T)\} \tag{1}$$
$$\text{BB}(T) = \{b \in \mathcal{N}(T) \mid \nexists c \text{ s.t. } (b, c) \in \mathcal{E}(T)\}.\} \tag{2}$$

### F.2  LOSS FUNCTION

Let the superscript $(i)$ indicate the $i$-th sample in the dataset. The loss function is:

$$\mathcal{L}_{\mathcal{D}}(\Phi) := \frac{1}{|\mathcal{D}|} \sum_{i=1}^{|\mathcal{D}|} \sum_{n \in \text{RXN}(T^{(i)})} \text{CE}(\pi_{\mathcal{R}}(T^{(i)}, \boldsymbol{H}^{(i)})_n, \boldsymbol{y}_n^{(i)}),$$

$$\mathcal{L}_{\mathcal{D}}(\Omega) := \frac{1}{|\mathcal{D}|} \sum_{i=1}^{|\mathcal{D}|} \sum_{n \in \text{BB}(T^{(i)})} \text{MSE}(\pi_{\mathcal{B}}(T^{(i)}, \boldsymbol{H}^{(i)})_n, \boldsymbol{y}_n^{(i)}).$$

CE and MSE denote the standard cross entropy loss and mean squared error loss, respectively. For our evaluation metric, we consider accuracy, where the output of $\pi_{\mathcal{B}}$ is interpreted as the nearest building block with respect to cosine distance.

### F.3  AUXILIARY TRAINING TASK

In Section 3.3.1, we defined the representation $T$ to be the parse tree of a partial program. However, we omitted an extra step that was used to preprocess $T$ for training. The motivation for this extra step is discussed deeply in D.2. We add an additional step when preprocessing $\mathcal{D}$: For each $T$ in $\mathcal{D}$, for each node $r$ corresponding to a reaction, we add a new node $o_r$ corresponding to the intermediate

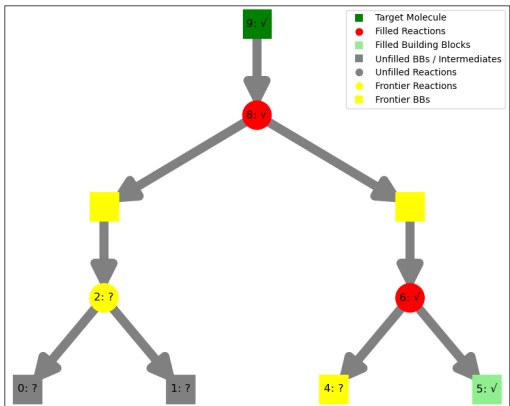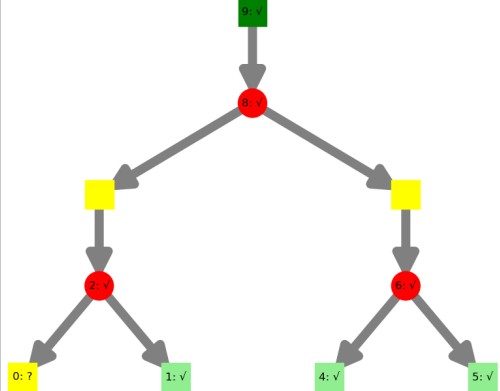

Figure 12: Examples of $T'$ where prediction targets are the frontier reactions (yellow circles), frontier building blocks (numbered yellow squares) and auxiliary intermediates (un-numbered yellow squares).

outcome of the reaction. If $\text{RXN}(T)$ is the reaction nodes of $T$, we can construct $T'$ from $T$ as follows:

$$\mathcal{N}(T') \leftarrow \mathcal{N}(T) \cup \text{RXN}(T) \tag{3}$$

$$\mathcal{E}(T') \leftarrow \mathcal{E}(T) \cup \{(\text{parent}(r), o_r), (o_r, r) \forall r \in \text{RXN}(T)\} \tag{4}$$

Lastly, we attribute each $o_r$ with the intermediate obtained from the original synthetic tree, i.e. executing the output of the program rooted at $r$. We featurize $\{y_o := \text{FP}_{256}(o_{\text{SMILES}})\}$ and add them as additional prediction targets to $\mathcal{D}$. Examples of $T'$ are given in Figure 12.

### F.4 ABLATION STUDY: AUXILIARY TASK

To understand whether the two key design choices for $\partial \mathcal{P}'$ are justified, we did two ablations:

1. We use the original description of $\partial \mathcal{P}$ in Section 3.3.1, i.e. without the auxiliary task.

2. We use $\partial \mathcal{P}'$, but without attributing the intermediate nodes (so the set of targets is the same as Ablation 1.)

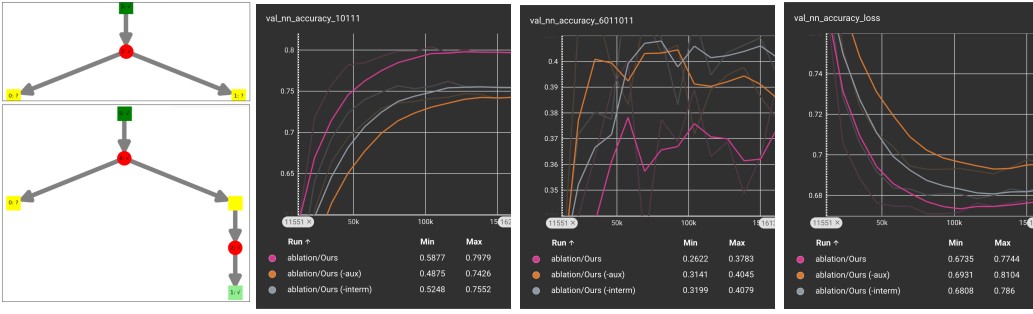

(a) Examples from $\partial \mathcal{P}'$    (b) NN accuracy loss over top example Figure 15a    (c) NN accuracy over bottom example Figure 15a    (d) NN accuracy over the validation set

Figure 13: We compare the proposed ablations on the NN accuracy metric over the whole dataset as well as on two specific syntactic classes.

As shown in Figure 13d, using $\partial \mathcal{P}'$ (Ours) achieves higher NN accuracy. This shows the benefit of learning the auxiliary training task. Meanwhile, ablating the auxiliary task (-aux) and ablating the intermediate node (-interm) does not have meaningful difference, indicating our architecture is

robust to graph edits which are semantically equivalent. To understand the comparative advantage vs disadvantage of the auxiliary training task, consider the two examples in Figure 15a. The first example is equivalent to learning a single-step backward reaction prediction on *forward* templates[3]. Our model clearly benefits from the auxiliary training task, which provides additional examples for learning the backward reaction steps. However, our model fares worse on predicting the first reactant of the top reaction. This may be due to competing resources. Despite the task being the same (and the set of forward templates are fixed), the model has to allocate sufficient capacity for the auxiliary task, whose output domain is much higher dimensional than $\mathcal{B}$. Ensuring positive transfer from learning the auxiliary task is an interesting extension for future work.

## F.5 ABLATION STUDY: LINEAR TRAINING

Table 6: We follow the same setup as Table 1, but retrain Ours using a dataset constructed with parameter $k = 4$ (whereas we used $k = 3$ in Table 1) to match the linear sampling strategy models, which we refer to as Ours:MC. We evaluate models trained using sampling constants 1 and 10, as described in 4.1.3.

| Dataset | Method | RR ↑ | Avg. Sim. ↑ | | | SA ↓ | | | Diversity ↑ | |
|---|---|---|---|---|---|---|---|---|---|---|
| | | | Top-1 | Top-3 | Top-5 | Top-1 | Top-3 | Top-5 | Top-3 | Top-5 |
| Test Set | Ours ($\tau$) | 56% | 0.827 | 0.633 | 0.555 | 3.100 | 3.019 | 2.918 | 0.543 | 0.628 |
| | Ours:MC1 ($\tau$) | 36% | 0.732 | 0.564 | 0.513 | 3.048 | 2.913 | 2.844 | 0.609 | 0.665 |
| | Ours:MC10 ($\tau$) | 65% | 0.869 | 0.658 | 0.609 | 3.163 | 3.000 | 2.928 | 0.558 | 0.610 |
| ChEMBL | Ours ($\tau$) | 7.6% | 0.531 | 0.443 | 0.396 | 2.544 | 2.510 | 2.460 | 0.675 | 0.727 |
| | Ours (MCMC) | 9.2% | 0.532 | 0.486 | 0.432 | 2.364 | 2.310 | 2.263 | 0.765 | 0.759 |
| | Ours:MC1 (MCMC) | 2.0% | 0.406 | 0.337 | 0.289 | 2.604 | 2.563 | 2.439 | 0.756 | 0.767 |
| | Ours:MC10 (MCMC) | 8.5% | 0.519 | 0.421 | 0.367 | 2.644 | 2.420 | 2.331 | 0.618 | 0.640 |

### F.5.1 RESULTS ON SYNTHESIZABLE ANALOG GENERATION

We study whether the efficiency of a sampling-based training strategy comes at a cost of performance. We make two observations from the results in Table 6.

**Constant factor matters.** The constant multiplier $C$ from $\mathcal{D}_0$ to $\mathcal{D}$ not only determines how much data the model sees each pass for the sake of efficiency. It is also be a parameter controlling the tradeoff between over-representing larger vs smaller templates. If it is larger than the largest number of masks (Table 5), the Linear strategy is essentially deactivated, since all masks will be used. At a lower value, only small programs with at most the number of masks as $C$ are fully represented. Medium-to-larger programs in $\mathcal{D}_0$ are under-represented, at the rate of the fraction of total masks that $C$ constitutes for its template class.

**Performance boost in-distribution.** We find that for $C = 10$, the performance is *better* across reconstruction, similarity and diversity with comparable SA to the standard training. Meanwhile for $C = 1$, the performance declines sharply. It is likely that standard training is overfitting to masks from larger programs, resulting in poorer generalization. Meanwhile, $C = 10$ downsamples those programs, and its sampling can be viewed as data-level regularization against overfitting.

**Slight performance drop out-of-distribution.** We find both $C = 1$ and $C = 10$ underperform compared to standard training. For $C = 10$, reconstruction and Top-1 similarity are actually comparable, but its similarity, SA and diversity are noticeably worse than standard training. Since ChEMBL feature predominantly unsynthesizable molecules, it is likely that the distribution of molecular fingerprints better reflect those outputs of the more complex programs in $\mathcal{D}_0$, which are downweighted by higher values of $C$.

### F.5.2 RESULTS ON SYNTHESIZABLE MOLECULAR DESIGN

**Depends on the task.** We also include preliminary results on synthesizable molecular design. We find the results hold up for the MC models. Encouragingly, for the difficult task of Osimertinib MPO,

---

[3]For some templates, the forward template is one-to-one. For others, applying the backward template results in an ill-defined precursor, due to the many-to-one characteristic of these templates.

Figure 14: We select the first Oracle from each Table in App. H to compare Ours with Ours (EP). Aside from the ablation networks, we use the same experimental settings as Table 2.

(a)

| | | | GSK3β | | | | | | | | | Median 1 | | | | | | | |
| | category | Oracle Calls | Top 1 | | | Top 10 | | | Top 100 | | | Oracle Calls | Top 1 | | | Top 10 | | | Top 100 | | |
| | | | Score | SA | AUC | Score | SA | AUC | Score | SA | AUC | | Score | SA | AUC | Score | SA | AUC | Score | SA | AUC |
|---|---|---|---|---|---|---|---|---|---|---|---|---|---|---|---|---|---|---|---|---|---|
| Ours (MC) | synthesis | 5056 | **0.98** | **2.045** | **0.923** | **0.97** | **2.294** | **0.893** | 0.942 | **2.294** | **0.814** | 7949 | **0.4** | 4.12 | 0.356 | **0.342** | **3.902** | 0.304 | 0.295 | **4.013** | **0.256** |
| Ours (MC10) | synthesis | | | | | | | | | | | 8045 | **0.4** | **3.353** | 0.357 | 0.344 | 4.593 | 0.297 | **0.302** | 4.44 | 0.247 |
| Ours | synthesis | 4886 | **0.98** | **2.045** | 0.891 | 0.967 | 2.302 | 0.848 | **0.944** | 2.27 | 0.778 | 8303 | **0.4** | **3.353** | **0.371** | **0.342** | 4.161 | **0.305** | 0.298 | 4.256 | 0.252 |

(b)

| | | | Osimertinib MPO | | | | | | | | | Perindopril MPO | | | | | | | |
| | category | Oracle Calls | Top 1 | | | Top 10 | | | Top 100 | | | Oracle Calls | Top 1 | | | Top 10 | | | Top 100 | | |
| | | | Score | SA | AUC | Score | SA | AUC | Score | SA | AUC | | Score | SA | AUC | Score | SA | AUC | Score | SA | AUC |
|---|---|---|---|---|---|---|---|---|---|---|---|---|---|---|---|---|---|---|---|---|---|
| Ours (MC) | synthesis | 9402 | 0.865 | 2.282 | 0.830 | 0.853 | **2.187** | 0.813 | 0.841 | 2.189 | 0.771 | 10000 | 0.572 | **3.101** | 0.541 | 0.567 | **3.072** | 0.521 | 0.555 | 3.077 | 0.486 |
| Ours (MC10) | synthesis | 5056 | **0.98** | **2.045** | **0.923** | **0.97** | 2.294 | **0.893** | **0.942** | **2.294** | **0.814** | 10000 | 0.596 | 3.263 | 0.542 | 0.574 | 3.147 | 0.523 | 0.556 | 3.058 | **0.489** |
| Ours | synthesis | 10000 | 0.859 | 2.263 | 0.826 | 0.847 | 2.21 | 0.81 | 0.832 | 2.249 | 0.769 | 10000 | **0.622** | 3.338 | **0.547** | **0.591** | 3.378 | **0.524** | **0.558** | **3.137** | 0.485 |

we see for $C = 10$, the results are substantially better. At the same time, Ours remain better for Perindopril MPO, which suggests each training strategy may suit different tasks. We also see for easier tasks like GSK and Median 1, the $C = 1$ model and Ours are essentially interchangeable. This suggests even a low reconstruction accuracy suffices for these Oracles.

**Implications.** The stratified sampling is designed to enhance supervised policy learning on larger programs while preventing the combinatorial explosion in Algo. 1 and to some extent, overfitting. However, this may benefit easier tasks (e.g. distributions of synthesizable molecules) while disadvantaging harder tasks. Thus, $C$ should be tuned for optimal trade-off between efficiency and downstream synthesizable analog generation performance.

### F.6    MODEL ARCHITECTURE

We opt for two Graph Neural Networks (for $\Phi, \Omega$), each with 5 modules. Each module uses a TransformerConv layer (Shi et al., 2020) (we use 8 attention heads), a ReLU activation, and a Dropout layer. We adopt sinusoidal positional embeddings via numbering nodes using the postorder traversal (to preserve the pairwise node relationships for the same skeleton). Then, we pretrain $\Phi, \Omega$ with $\mathcal{D}$.

### F.7    TRAINING & CONVERGENCE ANALYSIS

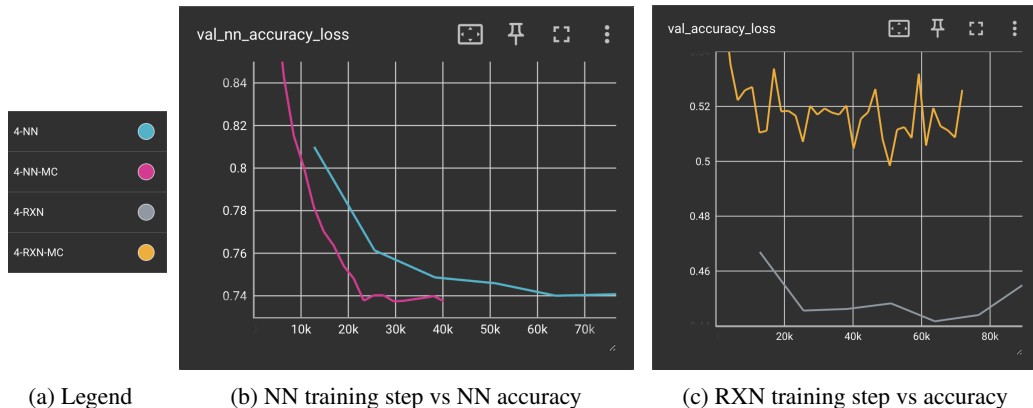

(a) Legend          (b) NN training step vs NN accuracy          (c) RXN training step vs accuracy

Figure 15: We plot the number of training steps needed to converge our models under the standard and linear (-MC) training strategies for $C = 1$.

We elaborate on 4.1.3 further with a quantitative comparison of training costs with SynNet Gao et al. (2022). The key difference is $\mathcal{D}_{\text{synnet}}$ batches by *the size of a synthetic tree*, whereas we batch by the synthetic trees (in program form).

**SynNet Training Complexity:** SynNet serializes the construction of a synthetic tree, so a training epoch does $O(\sum_{\mathcal{D}_0} |T|)$ passes, where $|T|$ is the number of nodes (or number of edges, as they are different by one). The cost of a MLP forward and backward pass for SynNet is

$O(L \cdot H^2 + F \cdot H)$. The total complexity per epoch is $O((\sum |T|)(L \cdot H^2 + F \cdot H))$.

**SynthesisNet Linear Training Complexity:** As discussed in 4.1.3, a training epoch does $O(\mathcal{D}_0)$ passes, where the constant factor can be adjusted. The cost of a GNN forward and backward pass on a tree $T$ is $O(L \cdot |T| \cdot (F \cdot H + H^2))$, where $L$, $F$, $H$ are number of layers, feature and hidden dimension, respectively. The total complexity per epoch is $O(L \cdot (\sum |T|) \cdot (F \cdot H + H^2))$ since we train on the forest of trees over $\mathcal{D}_0$. This is equivalent to $O((\sum |T|)(L \cdot H^2 + F \cdot H))$.

We can conclude the per-epoch complexity following the linear training strategy is equivalent to that of SynNet's. However, what matters in practice is the convergence rate, so we also include a quantitative comparison between convergence plots. We find that even SynthesisNet's standard training strategy is practically equivalent to SynNet in the number of passes needed to converge the model. As a disclaimer, we only include SynthesisNet's numbers using the default setting of $k = 4$, where $|\mathcal{D}_0| \approx 135k$ and $|\mathcal{D}| \approx 818k$. We also show results of linear training with constant factor of 1 (i.e. $|\mathcal{D}| = |\mathcal{D}_0|$).

**Convergence Comparison:** Both SynNet and SynthesisNet uses a batch size of 64. We see from Figure 15 that SynthesisNet requires $\approx 170k$ training steps (batches) to converge (combining both BB and RXN networks) following standard training. Meanwhile, we refer readers to Figure 13 of Gao et al. (2021), where the convergence plots show at least (with the most generous interpretation) 1M steps needed to converge *each* of the action, reactant1, reaction and reactant2 networks. Pooling the training of all networks, we give a *very* generous estimate of $1M$ batches for SynNet to converge. What's left is to figure the average scaling size factor from a batch of trees (ours) to a batch of tree nodes (SynNet), which computes as $818k/135k \approx 6$, which implies $\approx 6 \cdot 170k \approx 1M$ steps. We can conclude the SynthesisNet with *standard* training is comparable if not more efficient in the number of training steps to converge the model. Should efficiency be of further concern, we suggest using the Linear Training strategy, and adjusting the sample constant factor accordingly.

**Training and Inference Time:** Converging the RXN and BB networks took us $\approx 1$ and 5 hours on a single NVIDIA RTX A6000. A single inference call to the surrogate takes a few seconds.

### F.8   ATTENTION VISUALIZATION

We elucidate how our policy network leverages the full horizon of the MDP to dynamically adjust the propagation of information throughout the decoding process. Since our decoding algorithm decodes once for every topological order of the nodes, the actual attention dynamics can vary significantly. Thus, we show a prototypical decoding order where:

1. All reactions are decoded before building blocks.

2. If decoding a reaction, the reaction node which $\pi_{\mathcal{R}}$ predicts with the highest probability is decoded.

3. If decoding a building block, the node where the embedding from $\pi_{\mathcal{B}}$ has minimal distance to a building block is decoded.

In (Shi et al., 2020), each TransformerConv layer $l$ produces an attention weight for each edge, $[\alpha_{i,j}^{(l)}]$ where $\sum_j \alpha_{i,j}^{(l)} = 1$. We average over all layers to obtain the mean attention weight for each directed edge, i.e., we set the thickness of each edge $(i, j)$ in each subfigure of Figure 16 to be proportional $\sum_l \alpha_{i,j}^{(l)}$.

We make some generation observations:

- The information flow along child-parent edges indicate usage of the full horizon. This is the main feature of our approach compared to traditional search methods like retrosynthesis.

- Our positional embeddings enables asymmetric modeling. SMIRKS templates specify the order of reactants and is usually not arbitrary. We observe that more often than not, the parent attends to its left child more than the right child. This may be a consequence of template definition conventions, where the first reactant is the major precursor. The subtree under the node more likely to be the major precursor is more important for predicting the reaction.

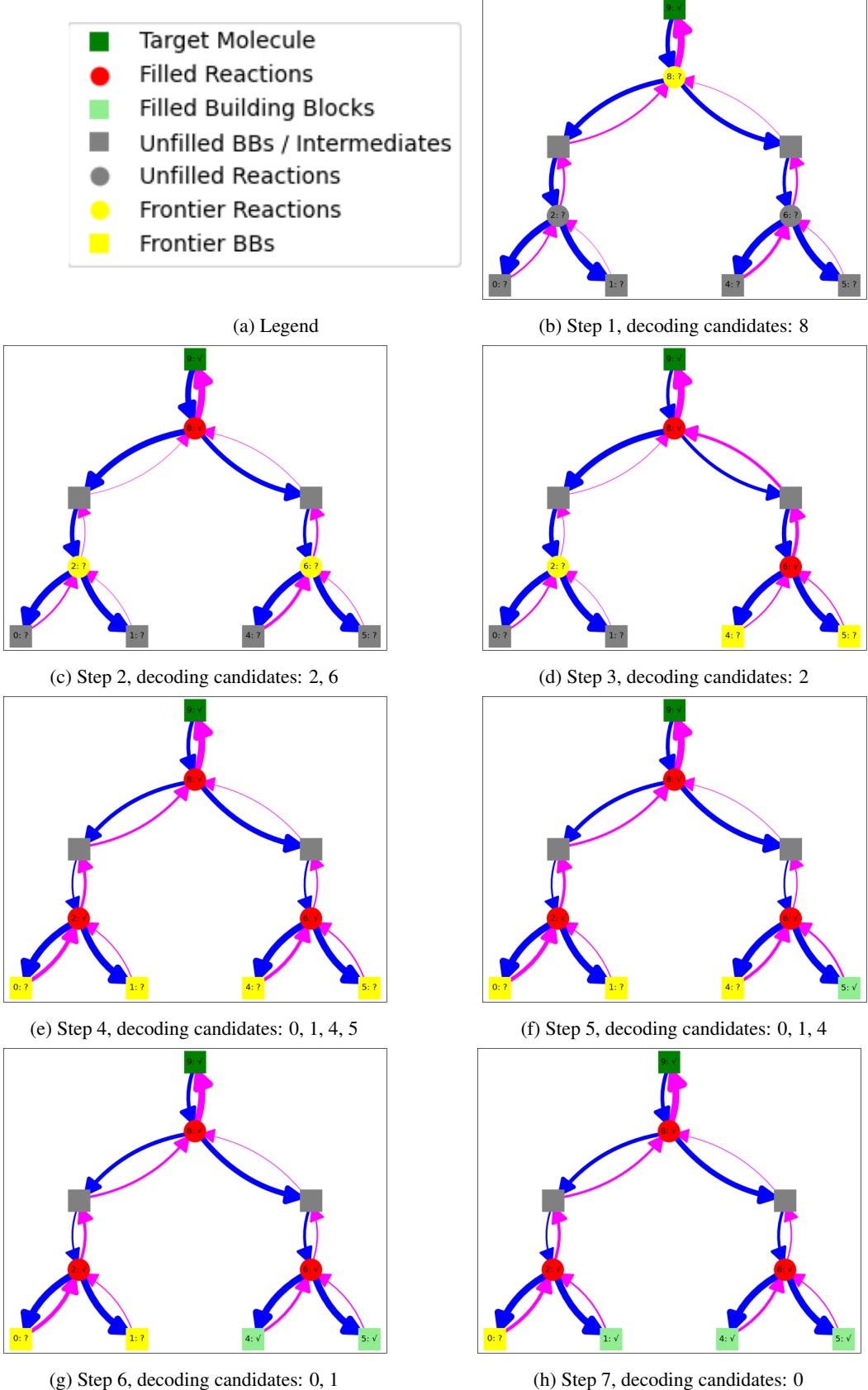

Figure 16: Case Study of Attention Flow

Now, we do a detailed walkthrough the 7-step decoding process to understand the evolution of the information flow. Each subfigure corresponds to the state of the MDP after a number of decoding steps, with the candidates of decoding colored in yellow. The attention scores are computed during the inference of $\Phi$ or $\Omega$ and averaged.

1. In Figure 16b, we see that 8 attends significantly to the target, unsurprisingly. 8 also attends to both its children, and attends more to its left child, which is a prior consistent with our general observation.

2. In Figure 16c, we see that after a specific reaction is instantiated at 8, the attention dynamics somewhat change. The edge from 8 to its left child thickens, while the edge from the left child to 8 thins. This is likely because now that the identity of 8 is known, it no longer needs to attend to its left child. The reciprocal relationship now intensifies, as the first reactant of 8 now attends to 8.

3. In Figure 16d, after the reaction at 6 is decoded, we see the information propagate back up the tree and to the other subtree to inform 2. We see the edge along the path from $6 - 8$ thickens, indicating the representation of 8 is informed with new information, and in turn propagates it to 2.

4. In Figure 16e, after the reaction at 2 is decoded, we see the same phenomenon happen, where the information flow again propagates back up and to the other subtree. However, we see this comes with a tradeoff, as 6 attends to its parent less, and instead reverts to its original attention strength to its children. We hypothesize the identity of 2 has a strong effect on the posterior of 6. This is an example where branching out to try more possible orders of decoding would facilitate a more complete algorithm.

5. In Figure 16f, we see how determining 5 causes 6 to attend more to 5 than it does to its parent. Knowledge of 5 allows the explaining away of 4.

6. In Figure 16g, we note instances of a general phenomenon: the second reactant is decoded followed by the first. Empirically, the distribution of the second reactant has lower entropy than the first. 4 was inferred after 5 as the knowledge of its parent reaction and sibling reactant likely constrains its posterior significantly.

7. In Figure 16h, we see a similar phenomenon where the representation of 2 attends slightly more to 1 after it is decoded.

In summary, the syntax structure of the full horizon is crucial during the decoding process. The attention scores allow us to visualize the dynamic propagation of information as nodes are decoded. Our observations highlights the flexibility of this approach compared to an infinite horizon formulation.

Table 7: Hyperparameters of our GA.

| | Parameter | Value |
|---|---|---|
| General | Max. generations | 200 |
| | Population size | 128 |
| | Offspring size | 512 |
| | Seed initialization | random |
| | Fingerprint size | 2048 |
| | Early stopping warmup | 30 |
| | Early stopping patience | 10 |
| | Early stopping $\Delta$ | 0.01 |
| Semantic evolution | Parent selection prob. of $i$ | $\propto (\mathrm{rank}(i) + 10)$ |
| | Num. crossover bits $n_{\mathrm{cross}}$ | $\mathcal{N}(1024, 205)$ |
| | Num. mutate bits $n_{\mathrm{flip}}$ | 12 |
| | Prob. mutate bits $p_{\mathrm{flip}}$ | 0.5 |
| Syntactic mutation | Num. tree edits $n_{\mathrm{edit}}$ | $\mathcal{U}\{1, 2, 3\}$ |
| Surrogate | Max. topological orders | 5 |
| | Sampling strategy | greedy |

# G  GENETIC ALGORITHM

Our genetic algorithm (GA) is designed to mimic SynNet's (Gao et al., 2021), and settings are given in Table 7. We fix the same number of offspring fitness evaluations per generation to ensure a fair comparison, strategically allocating the evaluations between offspring generated using semantic evolution and those generated using syntactic mutation.

## G.1  SEMANTIC EVOLUTION

Given two parents $x_1$ and $x_2$, semantic evolution samples a child $x_*$ as follows. We combine $n_{\text{cross}}$ random bits from $x_1$ and the other $2048 - n_{\text{cross}}$ bits from $x_2$ and then, with probability $p_{\text{flip}}$, flipping $n_{\text{flip}}$ random bits of the crossover result.

## G.2  SYNTACTIC MUTATION

Given a child $x_*$ from Appendix G.1, syntactic mutation performs $n_{\text{edit}}$ edits on $T = \tau(x_*)$ to obtain a syntactic analog $T$. With equal probability, each edit either adds or removes a random leaf. To do so, we enumerate all possible additions and removals, and ignore the ones that produce an empty tree or a tree with more than 4 reaction nodes. The edit is uniformly sampled from all such choices, or no operation is performed if no viable choices exist. Using the surrogate, the siblings $(x_*, T)$ and $(x_*, T)$ are then turned into two fingerprints, and one of with the higher expected improvement under a Gaussian process (GP) is selected. Our GP uses a radial basis function kernel with length scale 1 and is fitted using the population and offspring from the preceding generation.

## G.3  SURROGATE CHECKPOINT

The surrogate checkpoint was trained as described in Appendix F. To lower the runtime of the GA, we only reconstruct using a random subset of the input skeleton's possible topological orders. For each topological order, we follow a greedy decoding scheme where reactions are decoded before building blocks, as described in Appendix F.8.

# H  FULL RESULTS ON TDC ORACLES

Table 8: Guacamol structural target-directed benchmarks: Median 1 & 2 (average similarity to multiple molecules) and Celecoxib Rediscovery (hit expansion around Celecoxib).

Table 9: Bioactivity Oracles for GSK3B, JNK3, and DRD2

Tables 8, 9, 10 and 11 are comprehensive results against baselines taxonomized in (Gao et al., 2022). We evaluate the average score of the Top K molecules, their average synthetic accessibility (Ertl & Schuffenhauer, 2009) and top K AUC (AUC of no. oracle calls vs score plot), for K=1,10,100. Like

Table 10: Guacamol multi-objective Oracles for properties of known drugs: Osimertinib, Fexofenadine, Ranolazine.

Table 11: Guacamol multi-objective Oracles for properties of known drugs: Perindopril, Amlodipine, Sitagliptin, and Zaleplon. Disclaimer: We used version 1.0.0. PMO Gao et al. (2022), where we adopted the numbers from, used 0.3.6. We'd like to mention there were changes since 0.3.6 that may affect the Sitagliptin and Zaleplon evaluations.

(Gao et al., 2022), we limit to 10000 Oracle calls, truncating and padding to 10000 if convergence occurs before 10000 calls. For each cell, numbers are followed by rankings. $X(R_1|R_2)$ means score $X$ is ranked $R_1$-best amongst all methods for that column and $R_2$-best amongst in-category methods. We visualize the rankings in Figure 17 to facilitate easier interpretation of the results.

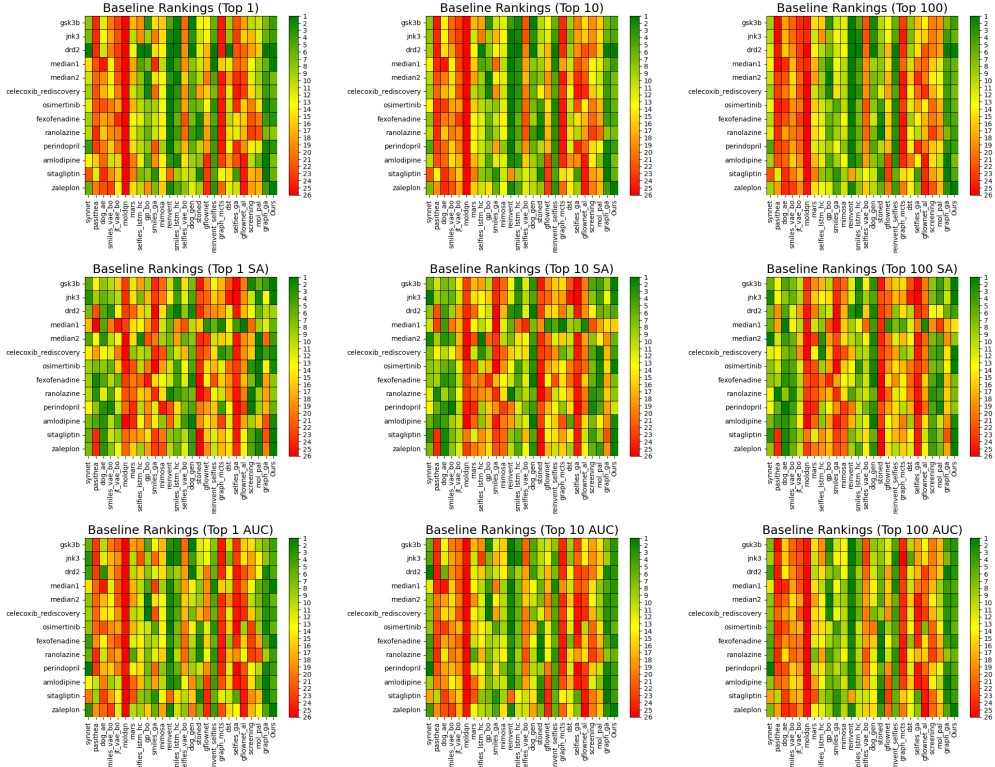

Figure 17: Ranking of our method against baselines on Top $k$ Average Scores (top), SA Scores (middle) and AUC (bottom), for $k = 1, 10, 100$ (left, middle, right).

# I   DOCKING CASE STUDY WITH AUTODOCK VINA

(a) Top 3 molecules with lowest binding energy against DRD3 and M$^{pro}$ from Ours vs SynNet

(b) Top binders against M$^{pro}$ from literature, based on consensus docking scores (Ghahremanpour et al., 2020)

We structurally analyze the top molecules discovered by our method, visualized in Figure 18a.

For our optimized binders against DRD3, the chlorine substituent and polycyclic aromatic structure suggest good potential for binding through $\pi - \pi$ interactions and halogen bonding. The bromine and carboxyl groups can enhance binding affinity through halogen bonding and hydrogen bonding, respectively. The polycyclic structure further supports $\pi - \pi$ stacking interactions. In general, they have a comparable binding capability to the baseline molecules, but with simpler structures, so the ease of synthesis for the predicted molecules are higher than the baseline molecules.

For our optimized binders against Mpro, the three predicted molecules contain multiple aromatic rings in conjugation with halide groups. The conformation structures of the multiple aligned aromatic rings play a significant role in docking and achieve ideal molecular pose and binding affinity to Mpro, compared to the baseline molecules shown in Figure 18b. The predicted structures indicate stronger $\pi - \pi$ interaction and halogen bonding compared with the baselines. In terms of ease of synthesis, Bromination reactions are typically straightforward, but multiple fused aromatic rings can take several steps to achieve. In general, the second and third can be easier to synthesize than the top binder due to less aromatic rings performed. However, the literature molecules appeared to be even harder to synthesize due to their high complexity structures. So the predicted molecules obtained a general higher ease of synthesis than the baseline molecules. Compared with the other baseline molecules, e.g. Manidipine, Lercanidipine, Efonidipine (Dihydropyridines), known for their calcium channel blocking activity, but not specifically protease inhibitors, Azelastine, Cinnoxicam, Idarubicin vary widely in their primary activities, not specifically designed for protease inhibition. Talampicillin and Lapatinib are also primarily designed for other mechanisms of action. Boceprevir, Nelfinavir, Indinavir, on the other hand, are known protease inhibitors with structures optimized for binding to protease active sites, so can serve as strong benchmarks. Overall, the binding effectiveness of the predicted molecules are quite comparable to the baseline molecules.