# OpenReview forum: "Procedural Synthesis of Synthesizable Molecules"
_ICLR.cc/2025/Conference — ICLR 2025 Poster_

### Official Review · Reviewer_iU5H · 2024-10-27

**Soundness:** 3
**Presentation:** 3
**Contribution:** 3
**Rating:** 6
**Confidence:** 4

**Summary:**

In this paper, the authors address the problems of synthesizable analog generation and synthesizable molecule design, employing syntax-guided synthesis techniques from the field of program synthesis. Specifically, they developed a bi-level framework that decouples the syntactical skeleton of a synthetic tree from the program semantics, and introduced amortized RL algorithms based on the framework. They demonstrated improvements across multiple dimensions of performance for both tasks, and include in-depth visualizations and ablation studies.

**Strengths:**

1. The problem of designing useful molecules with desirable properties and good synthesizability is very critical to drug discovery.
2. The connection between the problem and the program synthesis community is vital.
3. The method is carefully designed and well executed in experiments.

**Weaknesses:**

1. Since this work heavily involves discussion on search space, and amortized search within the synthesis tree, similar things are also explored deeply in the retrosythesis community. Thus, it would be beneficial to incorporate some discussions in the paper to discuss the connections and differences between this work and previous work.

    1. Self-Improved Retrosynthetic Planning, Kim et al., ICML 2021
    2. Retrosynthetic Planning with Dual Value Networks, Liu et al., ICML 2023
    3. Retrosynthesis Zero: Self-Improving Global Synthesis Planning Using Reinforcement Learning, Guo et al., JCTC 2024


2. The paper is clear and well-written. However, it would be helpful if the introduction includes a reference to the detailed description of the two tasks in the related work section. This will help readers understand the tasks before they delve into the techniques used.

3. Regarding the experimental setup, could you explain the choice of the 91 reaction templates and the 147,505 building block compounds? Are they forward reaction templates? Do these choices reflect real-world applications in molecular design?

**Questions:**

See above

---

> ### Author Response · Authors · 2024-11-28
> **Rebuttal by Authors**
>
> **Thank you so much for acknowledging the importance, vitality and careful execution of our work!**
>
> 1. *Since this work heavily involves discussion on search space, and amortized search within the synthesis tree, similar things are also explored deeply in the retrosythesis community. Thus, it would be beneficial to incorporate some discussions in the paper to discuss the connections and differences between this work and previous work.*
>
> - *Self-Improved Retrosynthetic Planning, Kim et al., ICML 2021*
>
> - *Retrosynthetic Planning with Dual Value Networks, Liu et al., ICML 2023*
>
> - *Retrosynthesis Zero: Self-Improving Global Synthesis Planning Using Reinforcement Learning, Guo et al., JCTC 2024*
>
> Thank you for pointing us to these interesting works! We have read these works and included a thorough discussion of the similarities and differences in App. D.3. We are familiar with and acknowledge that the retrosynthesis community has adopted sophisticated techniques like iterative policy improvement under self-play and we included discussion of potential ways our work can incorporate them in the next iteration. We also included a deeper dive into how the difference between forward and retrosynthesis informed the development of our method in App. D.2. We hope you find the discussions interesting!
>
> 2. *The paper is clear and well-written. However, it would be helpful if the introduction includes a reference to the detailed description of the two tasks in the related work section. This will help readers understand the tasks before they delve into the techniques used.*
>
> Thank you for this helpful suggestion! We added a few sentences in the Introduction with references to prior works that describe the tasks more precisely. We also modified certain descriptions, e.g. “synthetic pathways” => “synthetic procedures” to avoid ambiguity with the retrosynthesis problem (since sometimes “pathways” is used to describe backward templates). We’ve also expanded Section 3.1 to define the problems more precisely for our program synthesis based formulation.
>
> 3. *Regarding the experimental setup, could you explain the choice of the 91 reaction templates and the 147,505 building block compounds? Are they forward reaction templates? Do these choices reflect real-world applications in molecular design?*
>
> Yes, yes, and yes!
>
> The building block compounds come directly from Enamine, a supplier of purchasable compounds (https://enamine.net/). These building blocks are used to construct make-on-demand libraries by Enamine REAL (REadily AccessibLe) Space [1], so they are reliable starting materials for robust synthesis protocols. The REAL Space has an average synthesis rate of 80% for 1-3 synthesis steps, attesting to the validity of the building blocks.
>
> The 91 reaction templates are curated by Button & Hartenfeller and were defined in close collaboration with experts. They include well-known organic transformations such as the Wittig reaction [2], Buchwald-Hartwig reaction [3], and various functional group conversions. By category, 13/91 are uni-molecular and 78 are bi-molecular reactions. 63 are skeleton formation, 23 for peripheral modifications and the remaining 5 used for either. For the skeleton formation reactions, 45 are ring formation reactions, with 37 heterocycle formations and 8 carbocycles. These templates seek to codify the most robust “rules of chemistry” and approximate real laboratory organic synthesis procedures. They can be found under data/assets/reaction-templates/hb.txt of our anonymous repo, which we just made accessible. The templates are written in SMARTS. You can visualize them with (https://smarts.plus). As a fun sidenote, you can prompt ChatGPT with the template, asking it to describe what it does, and it does a good job describing the reaction and its common use cases. We can contrast this set with the much larger (~380K) set of retrosynthesis templates derived from retrosynthesis databases like USPTO. They can be automatically extracted using tools like RDChiral and complement search algorithms, but they have not been individually vetted to be robust chemical transformations that serve real-world applications. In the (moved) App. D.2., we explain why this matters for developing our methodology.
>
> [1] Grygorenko, Oleksandr O., et al. "Generating multibillion chemical space of readily accessible screening compounds." Iscience 23.11 (2020).
>
> [2] Wittig, Georg, and Georg Geissler. "Zur Reaktionsweise des Pentaphenyl‐phosphors und einiger Derivate." Justus Liebigs Annalen der Chemie 580.1 (1953): 44-57.
>
> [3] John F Hartwig. Transition metal catalyzed synthesis of arylamines and aryl ethers from aryl halides and triflates: Scope and mechanism. Angewandte Chemie International Edition, 37(15):2046–2067, 1998.

---

> > ### Author Response · Authors · 2024-12-02
> > **Looking forward to your response**
> >
> > Dear reviewer,
> >
> > Thank you again for your interesting suggestions and thoughtful consideration! We’ve worked hard to thoroughly address your suggestions and further refine the paper, aiming to make it the best it can be. We’d greatly appreciate your feedback on whether your concerns have been addressed!
> >
> > Sincerely,
> >
> > Authors

---

### Official Review · Reviewer_Z3YQ · 2024-11-10

**Soundness:** 3
**Presentation:** 4
**Contribution:** 3
**Rating:** 8
**Confidence:** 3

**Summary:**

This paper presents a method for synthesize of synthesizable molecules. It includes separates the syntactic skeleton (structure) of a molecule from the semantics (functional groups and properties), Markov Chain Monte Carlo (MCMC) Simulations for skeleton refinement, Genetic Algorithms for the optimization of both the structural and chemical aspects of molecules, and Surrogate Modeling with Graph Neural Networks to represent the molecular structure.

The experiments show that the proposed framework matches and outperforms the current state of the art approaches to synthesizable molecule and synthesizable analog design.

**Strengths:**

- The approach is interesting from the soft-computing point of view. The authors leverage the four different approaches on the right places. The separation between the structure and the content in the synthesis approach is interesting in particular. The use of MCMC and GA is more standard but is well suited for the new candidate tree generation and the search of the best structure and content of candidate molecule.

**Weaknesses:**

- I think the possible weakness is the dependency on the tree and the grammar components. On one hand having a very large amount of templates will increase the computational complexity of the model (it is not clear how for instance the MCMC algorithm would handle this) and on the other hand a more efficient smaller set will not allow to generate all desired solutions

**Questions:**

How would the method handle a much larger set of the input alphabet and structural templates?

---

> ### Author Response · Authors · 2024-11-28
> **Rebuttal by Authors**
>
> **Thank you so much for acknowledging the innovation and excellence of our approach!**
>
> 1. *I think the possible weakness is the dependency on the tree and the grammar components. On one hand having a very large amount of templates will increase the computational complexity of the model (it is not clear how for instance the MCMC algorithm would handle this) and on the other hand a more efficient smaller set will not allow to generate all desired solutions*
>
> Thanks for raising these points! We study whether a larger set of templates can be incorporated by studying whether our framework **extrapolates** to unseen templates at inference time. Note this argument relies on the observation that whether the method can handle a much larger template set at test time is reducible to whether the method can extrapolate on any given new, unseen template. We’ve added new sections (4.3.3 and App. B) for a new ablation study. Specifically, we hold out 25% of templates, retrain our model, and see if our surrogate model is still performant. In particular, we set up our study so all original templates can still be used at inference time (they are only excluded from training). This will test whether 1) our method can incorporate unseen and potentially a much larger set of templates for downstream purposes, and 2) whether this matters for the final results. Here are some takeaways from the study:
> - Our ablation surrogate model can reconstruct programs from templates it has not seen before to a comparable level of accuracy as the original model.
> - Similarity and diversity of generated analogs only drops marginally on the Test Set. What’s surprising is that performance actually held up on the 25% set of templates which were held out from training. We suspect removing the templates induces a regularization effect, implying the model is not underfitting to those templates.
> - Performance drops slightly when tested with the ChEMBL, which comprises mostly unsynthesizable molecules. Diversity decreases, to some extent confirming your suspicion.
> - Interestingly, the downstream molecular design results are not hurt at all, and in fact sample efficiency is improved in some cases. This suggests the ablation model can still generate sufficiently many analogs within our bilevel optimization framework.
>
> Also, an alternative to MCMC we propose for navigating a large number of templates is to “recognize” the template with a learned predictor (introduced in Section 3.3). Although efficient, we show it maintains respectable analog generation performance for both downstream tasks, as a baseline (Table 1) and an ablation (Table 4).
>
> Also, we added new sections 4.1.3, Appendix F5 and Appendix F7 to discuss the computational complexity and its dependency on the grammar, as well as a new sampling strategy to mitigate the complexity explosion from using larger templates.
>
> (continued in next reply)

---

> > ### Author Response · Authors · 2024-11-28
> > **Rebuttal by Authors**
> >
> > 1. *How would the method handle a much larger set of the input alphabet and structural templates?*
> >
> > Thanks for this question! To handle a much larger set of input alphabet, let’s look at both building blocks (literals) and reactions (operators) separately.
> >
> > For building blocks, the solution (which we already implemented) is to amortize over the number of building blocks by using techniques from metric learning. We use Morgan Fingerprints [1] to embed building blocks as fixed dimension embeddings and build an index for nearest neighbors. Instead of predicting over all the (>100K) classes, we can predict the embedding and use SOTA packages for nearest neighbor retrieval at scale (such as faiss).
> >
> > For reactions, this is less of a concern since we currently use only 91 robust, expert-codified forward reaction templates. They reflect common real-world reactions and are designed to approximate organic transformations done in real-world laboratories. You can refer to our discussion with iU5H for further details. However, if we had used a much larger library of reaction templates, e.g. automatically extracted from reaction databases like USPTO, we would similarly try to embed them in fixed-dimension representations. Reaction templates are actually just a mapping between sub-molecular patterns (reactants to products), and each pattern can be embedded using Morgan Fingerprint. However, the atom-level mapping from reactants to products (where atoms are numbered and the numbering is respected during the mapping) requires further care. A newer, less-explored approach is to adopt a graph-based representation to represent the semantics of this transformation, where additional edges connect corresponding atoms between the reactants and products, achieving a 1:1 correspondence between the reaction template and this graph. We can then write an algorithm similar to how Morgan Fingerprints are computed to embed this reaction, based on neighborhood aggregation and hashing, the details of which we will leave for future iterations of our work.
> >
> > Thanks again for your incredible review and we hope we’ve addressed your concerns!
> >
> > [1] Morgan, Harry L. "The generation of a unique machine description for chemical structures-a technique developed at chemical abstracts service." Journal of chemical documentation 5.2 (1965): 107-113.

---

> > > ### Author Response · Authors · 2024-12-02
> > > **Looking forward to your response**
> > >
> > > Dear reviewer,
> > >
> > > Thank you again for your thoughtful and positive evaluation of our work. We’d greatly appreciate your feedback on whether your concerns have been fully addressed!
> > >
> > > Sincerely,
> > >
> > > Authors

---

### Official Review · Reviewer_XGkL · 2024-11-11

**Soundness:** 3
**Presentation:** 2
**Contribution:** 3
**Rating:** 6
**Confidence:** 4

**Summary:**

The paper reframes molecule design and synthesizable analog recommendation as conditional program synthesis problems. It introduces a bi-level framework that separates the syntactic skeleton of synthesis pathways from their chemical semantics, allowing efficient exploration of both syntactic and semantic design spaces using evolutionary algorithms and Markov Chain Monte Carlo simulations. By leveraging fixed-horizon Markov decision processes, the approach improves synthesizable molecule generation and offers control over synthesis complexity. Results demonstrate enhanced performance and resource efficiency, positioning this method as a promising tool for automated molecular discovery.

**Strengths:**

1. Frames molecular design and synthesizable analog recommendation as conditional program synthesis tasks, offering a fresh perspective in this field.
2. Demonstrates robust performance across key metrics, underscoring the effectiveness of the proposed methods.
3. Provides thorough experiments that validate the approach and its contributions to molecular design and synthesis.

**Weaknesses:**

1. The current approach uses a limited number of templates, and it is unclear how this framework could be expanded to include a broader range of templates, which could limit its flexibility.
2. Although the authors claim efficiency, the paper lacks direct comparisons to demonstrate this advantage against other methods.
3. The comparison between tasks in Section 3.1 could be enhanced with mathematical notation alongside chemistry examples. While the method draws on program synthesis concepts, the explanation may be confusing without using clear chemical illustrations.

**Questions:**

The primary concern lies in the method's scalability under the current scheme. It remains unclear how the approach would handle a significantly larger design space with more templates, and how its performance would hold up as template diversity increases.

---

> ### Author Response · Authors · 2024-11-28
> **Rebuttal by Authors**
>
> **Thank you for recognizing the novelty, effectiveness and validity of our work!**
>
>
> 1. *The current approach uses a limited number of templates, and it is unclear how this framework could be expanded to include a broader range of templates, which could limit its flexibility.*
>
> Thank you for bringing up this essential question about our method. A large-scale template set expansion is the natural next step following this initial pilot framework. Before going there, we answer a key question first: does our framework extrapolate well to a broader range of templates at test time? If the answer is yes, we can expand the framework to handle a broader range of templates, demonstrating the flexibility of our method. We added new sections Section 4.3.3 and Appendix B to answer that question. Specifically, we hold out 25% of templates, retrain our model, and see if our surrogate model is still performant. Some findings include:
> - Our ablation surrogate model can reconstruct programs from templates it has not seen before to a comparable level of accuracy as the original model.
> - Removing a fraction of the templates and all program occurrences from the dataset does not detriment downstream analog generation performance by much.
> - We actually see marginal improvements in sample-efficiency for synthesizable molecular design, while upholding similar overall Top k performance.
> The findings show the framework is flexible enough to include new templates while maintaining similar performance to if the framework had used all the templates during training. We hope this is a positive indication that our framework would be able to handle a much larger set of templates, provided enough similar templates are supplied for training. We leave the template set scaling laws as directions for future work.
>
> 2. *Although the authors claim efficiency, the paper lacks direct comparisons to demonstrate this advantage against other methods.*
>
> Thank you for bringing up another essential question about our method. We apologize for not including this in the initial submission. We hope the list of following additions in our Revision sufficiently address your concern:
> - Added 4.1.3: We describe the complexity of dataset construction and training and explain why despite the seemingly high complexity, it is still efficient in practice. We also introduce a new stratified sampling strategy designed to scale to larger templates. It has the desirable linear complexity and emits a tunable parameter C to trade-off efficiency with performance.
> - Revised App. A: We include statistics from our dataset to make the discussion around efficiency grounded in actual numbers.
> - Added App. F.5: We study the performance using the simplified training strategy across downstream tasks, observing a performance drop when C=1 but competitive performance when C=10, and discuss the implications.
> - Added App: F.7: We analyze the computational complexity under the simplified training strategy, observing equivalent big O complexity to SynNet. For C=1, we include convergence plots and do some back-of-the-envelope calculations to show the efficiency of our training. Interestingly, we also show convergence for the standard training is only slower by a factor of 1-2x compared to the simplified training strategy.
>
> 3. *The comparison between tasks in Section 3.1 could be enhanced with mathematical notation alongside chemistry examples. While the method draws on program synthesis concepts, the explanation may be confusing without using clear chemical illustrations.*
>
> We apologize for this omission from our initial manuscript. The revised version includes mathematical notation for the two tasks and elaborates on their mechanics. The description refers to Figure 1 for illustrations of the terminologies used. We hope this improves the readability of that Section. Due to current space limitations, we refer the reader to Figure 3 of a (cited) prior work for chemical illustrations. If this is not enough, let us know and we can do the best to move other parts of the main text around to incorporate our own chemical illustrations.
>
> *The primary concern lies in the method's scalability under the current scheme. It remains unclear how the approach would handle a significantly larger design space with more templates, and how its performance would hold up as template diversity increases.*
>
> Thanks for bringing this up again. We address these questions with our new study in 4.1.3 and address the implications in App. B. We note our argument relies on the following observation: whether the method can handle a much larger template set at test time is reducible to whether the method can extrapolate on any given new, unseen template. We hope our findings support the rest of our argument following this premise.
>
> Feel free to let us know if you have further suggestions or comments!

---

> > ### Author Response · Authors · 2024-12-02
> > **Looking forward to your response**
> >
> > Dear reviewer,
> >
> > Thank you again for your constructive comments! We’ve worked hard to thoroughly address your suggestions and further improve the paper, aiming to make it the best it can be. We’d greatly appreciate your feedback on whether your concerns have been addressed!
> >
> > Sincerely,
> >
> > Authors

---

### Official Review · Reviewer_EsMz · 2024-11-12

**Soundness:** 3
**Presentation:** 3
**Contribution:** 3
**Rating:** 6
**Confidence:** 4

**Summary:**

1. This paper addresses the challenge of designing synthesizable molecules and creating analogs for unsynthesizable ones by framing these tasks within a program synthesis framework. The paper introduces a bi-level approach to explore synthesis pathways by separating the structural skeleton of a synthetic pathway from its functional properties.
2. Through Markov Chain Monte Carlo simulations, they refine molecular structures iteratively and optimize both syntactic and semantic dimensions with evolutionary algorithms.

**Strengths:**

1. The problem statement is well-defined, and the methods for synthesizing analogs and generating molecules are clearly explained, including the program's semantics.
2. The method achieves state-of-the-art performance in molecule generation on benchmark datasets and demonstrates significantly greater efficiency than the SynNet method when tested with various oracles, such as GSK, JNK, and DRD2.
3. Experimental analysis was conducted using various evaluation metrics, including bioactivity predictors (oracles), structural profiles, multi-property objectives, and docking simulations. The extensive number of experiments, detailed in Appendix G, provides strong evidence that this method outperforms other benchmarking methods.
4. The figures in the paper effectively clarify the methodology. The t-SNE and MDS plots in Appendix B, based on data from the final hidden layer representation of the MLP, clearly illustrate the most popular skeleton classes.
5. The model architecture used in the method is thoroughly explained, with detailed insights provided in the attention visualization section in Appendix E.

**Weaknesses:**

1. This paper failed to mention the source code / anonymous repository and also in Appendix E . 6 ATTENTION VISUALIZATION figure number is missing.
2. Results are compared against the 2022 paper; The authors have not compared the results against any recent publications.
3. This paper doesn't address the computational cost or effectiveness of the algorithms. How long does it take to train the inner loop given ~136k synthetic trees, molecule generation or analog creation?

**Questions:**

1. How robust is bi-level framework to work with other architectures other than TransformerConv, something like Decision Transformer ?
2. Is there any recent papers like GFlowNet that could be referred other than synthesis-based SynNet published in 2022?

---

> ### Author Response · Authors · 2024-11-28
> **Rebuttal by Authors**
>
> **Thank you for recognizing the many strengths of our work!**
>
>
> 1. *This paper failed to mention the source code / anonymous repository and also in Appendix E . 6 ATTENTION VISUALIZATION figure number is missing.*
>
>
> [Here’s](https://anonymous.4open.science/r/SynthesisNet-6D4F/README.md) our anonymous repo! We’ve included detailed instructions, illustrations and commands in the README. We added the correct label to the figure -- nice catch!
>
>
> 2. *Results are compared against the 2022 paper; The authors have not compared the results against any recent publications.*
>
>
> Here are additional comparisons we did with ChemProjector [1] and SynFormer [2], recent follow-up works to the 2022 SynNet paper that propose Transformer-based solutions for synthesizable analog generation (ChemProjector, SynFormer) and synthesizable molecular optimization (SynFormer). Note that a fair comparison of SynthesisNet with them is tricky due to a few factors (that work to their favor):
>
> - The building block set is supplied by Enamine, which regularly updates their catalogues. Both works use the Oct. 2023 version which contains 211,220 molecules, but that set is not publicly released by them. Plus, that version has also been updated since. Meanwhile, we use the older version adopted by SynNet. Post discussion period, we will update the final (hopefully stronger) results on the newest catalogue data for the camera ready.
>
> - The reaction template set used by SynFormer is a different, larger set (115 templates) than the set used by SynNet, ChemProjector and Ours (91). They one-hot encode the templates, so it cannot be changed at inference time.
>
>
> We tried to resolve these factors as follows:
>
> - To standardize comparisons, we tried retraining those models on the building block and reaction sets used by SynNet and us. However, SynFormer reported doing distributed training across 8 A100 GPUs for over 6 days, spanning 730k training steps. Our work required nowhere close to that much…. For reference, our models were trained on a single NVIDIA RTX A6000 and required only a few hours. Unfortunately, we could not get our hands on that much resources so we cannot complete the retraining during the discussion period window.
> - Instead, we set up the following comparison. We use their released model checkpoints, but constrain their inference to the building block set used by us and SynNet. This has the following downstream effect:
>     - For ChemProjector, ignoring any positive (or negative, but unlikely) transfer from pretraining on a larger superset of building blocks, this is as close to a fair comparison as we can achieve.
>     - For SynFormer, the same applies, but they still get to use their larger set of 115 reaction templates.
>
> We will do the best to resolve these disparities by including fair comparisons with SynFormer and ChemProjector in the camera-ready version, but for now, here’s the preliminary results:
>
>
> **Results on Synthesizable Analog Generation (vs ChemProjector)**
>
>
> |         |               |               | Avg. Sim. $\uparrow$ |       |       | SA $\downarrow$ |       |       | Diversity $\uparrow$ |       |
> |---------|---------------|:-------------:|:--------------------:|:-----:|:-----:|:---------------:|:-----:|:-----:|:--------------------:|:-----:|
> | Dataset | Method        | RR $\uparrow$ |         Top-1        | Top-3 | Top-5 |      Top-1      | Top-3 | Top-5 |         Top-3        | Top-5 |
> | ChEMBL  | ChemProjector |     9.6\%     |         0.564        | 0.528 | 0.508 |      2.770      | 2.811 | 2.825 |         0.393        | 0.424 |
> |         | Ours (MCMC)   |     9.2\%     |         0.532        | 0.486 | 0.432 |      2.364      | 2.310 | 2.263 |         0.765        | 0.759 |
>
>
>
>
> We run ChemProjector’s default command for projecting molecules. Other than rebuilding the fingerprint index with the smaller building block set, all hyperparameters are the same. Although ChemProjector has the edge on reconstructive similarity, our method generates significantly more diverse (nearly 2x internal diversity) sets of analogs. Since the paper focuses on projecting to synthesizable space and evaluates only on hit expansion tasks, it may not consider the same desiderata for analog generation as us (e.g. localized edits vs structural diversity). However, a common desideratum that is crucial to consider is the simplicity / ease of synthesis, not just synthesizability. Our method controls this by design due to control over the structure of the synthetic pathway, and our MCMC framework biases towards simpler templates, and our results show up to 0.56 lower average SA score.
>
> (continued in next reply)

---

> > ### Author Response · Authors · 2024-11-28
> > **Rebuttal by Authors**
> >
> > **Results on Synthesizable Molecular Design (vs SynFormer)**
> >
> >
> > |           |           |  GSK3$\beta$ |       |       |       |        |       |       |         |       |       |   Median 1   |       |       |       |        |       |       |         |       |       |
> > |-----------|-----------|:------------:|-------|-------|-------|--------|-------|-------|---------|-------|-------|:------------:|-------|-------|-------|--------|-------|-------|---------|-------|-------|
> > |           |           |              | Top 1 |       |       | Top 10 |       |       | Top 100 |       |       |              | Top 1 |       |       | Top 10 |       |       | Top 100 |       |       |
> > |           | category  | Oracle Calls | Score |   SA  |  AUC  |  Score |   SA  |  AUC  |  Score  |   SA  |  AUC  | Oracle Calls | Score |   SA  |  AUC  |  Score |   SA  |  AUC  |  Score  |   SA  |  AUC  |
> > | SynFormer | synthesis |         5501 | 0.99  | 1.975 | 0.932 | 0.977  | 2.285 | 0.901 | 0.949   | 2.461 | 0.844 |         4771 | 0.4   | 3.353 | 0.35  | 0.321  | 3.774 | 0.292 | 0.281   | 3.843 | 0.252 |
> > | Ours      | synthesis |         4886 | 0.98  | 2.045 | 0.891 | 0.967  | 2.302 | 0.848 | 0.944   | 2.27  | 0.778 |         8303 | 0.4   | 3.353 | 0.371 | 0.342  | 4.161 | 0.305 | 0.298   | 4.256 | 0.252 |
> >
> >
> > |           |           | Perindopril MPO |       |       |       |        |       |       |         |       |       | Ranolazine MPO |       |       |       |        |       |       |         |       |       |
> > |-----------|-----------|:---------------:|:-----:|:-----:|-------|:------:|:-----:|-------|:-------:|:-----:|-------|----------------|-------|-------|-------|--------|-------|-------|---------|-------|-------|
> > |           |           |                 | Top 1 |       |       | Top 10 |       |       | Top 100 |       |       |                | Top 1 |       |       | Top 10 |       |       | Top 100 |       |       |
> > |           | category  |   Oracle Calls  | Score |   SA  |  AUC  |  Score |   SA  |  AUC  |  Score  |   SA  |  AUC  | Oracle Calls   | Score | SA    | AUC   | Score  | SA    | AUC   | Score   | SA    | AUC   |
> > | SynFormer | synthesis |       7112      | 0.582 | 4.498 | 0.545 |  0.558 | 4.477 | 0.517 |  0.530  | 3.978 | 0.481 | 6507           | 0.816 | 3.084 | 0.775 | 0.804  | 3.183 | 0.754 | 0.760   | 3.908 | 0.700 |
> > | Ours      | synthesis |      10000      | 0.859 | 2.263 | 0.826 | 0.847  | 2.21  | 0.81  | 0.832   | 2.249 | 0.769 |          10000 | 0.808 | 3.201 | 0.774 | 0.805  | 3.205 | 0.741 | 0.794   | 3.254 | 0.686 |
> >
> > We evaluate GraphGA-SF across a core set of Oracles: GSK3B, Median 1, Osimertinib MPO, Perindopril MPO (the first from each category in 4.1.4). However, we encountered a bug within their GA crossover operation when running Osimertinib MPO that didn’t trigger for the other three Oracles. The run terminates after around 200 Oracle calls. Instead,we tried to replace Osimertinib with another MPO Oracle, but after going through all of them, we realized the bug triggers across 5 of the 7 Oracles! Only the Perindopril and Ranolazine runs finished. We suspect there is some failure mode the authors didn’t address, which is understandable considering the SynFormer codebase was initially released only 2 months ago. We have contacted the authors about it. For now, we just replace Osimertinib with Ranolazine.
> >
> >
> > Across all 36 columns, our method is better or tied on 20 of them (vs SynFormer’s 19). Considering the GSK3B Oracle is a random forest predictor of bioactivity (where SynFormer scores 8 vs 1) which can be volatile at times, the comparison leans in our favor. However, we will still follow the steps outlined above to include a fair comparison against SynFormer ahead for the camera-ready version.
> >
> >
> > 3. *This paper doesn't address the computational cost or effectiveness of the algorithms. How long does it take to train the inner loop given ~136k synthetic trees, molecule generation or analog creation?*
> >
> > We’ve added detailed analysis on the computational complexity and practical training costs in Section 4.1.3 and Appendix F.7, referencing additional statistics we compiled in Appendix A. The TLDR is:
> > (F.7) We show the convergence rate of SynthesisNet is comparable (if not better) than SynNet, using plots and cost analysis to demonstrate so.
> > (4.1.3) We introduce an alternative “linear” training strategy and show it has linear complexity to $\mathcal{D}_0$ (the initial generated data) while still retaining positive support over all of $\mathcal{D}$ (our constructed dataset).
> > (F.7) We show our standard training is practically comparable to the linear training strategy, despite its theoretical complexity in 4.1.3. This can be accounted for by the statistics in Appendix A.
> >
> > (continued in next reply)

---

> > > ### Author Response · Authors · 2024-11-28
> > > **Rebuttal by Authors**
> > >
> > > 1. *How robust is bi-level framework to work with other architectures other than TransformerConv, something like Decision Transformer?*
> > >
> > > Thanks for the interesting suggestion! In our framework, TransformerConv can be replaced with any architecture that can process a tree and predict node embeddings. TransformerConv is chosen for its simplicity and usefulness in visualizing the local attention weights. Decision Transformer is more of a framework shift, in which we need to formulate an appropriate MDP first then cast it as a sequence modeling problem. In our MDP, each state is a partial program (partially filled skeleton), so embedding the state as a token may relinquish the useful inductive bias of its tree structure. However, a Transformer architecture can indeed be used instead of TransformerConv, enabling long-range dependence modeling, and a causal self-attention mask can replace the flexible decoding order of our work. We encourage follow-up works to explore this idea. We’ve added this discussion (along with discussion of GFlowNet) to Appendix D of the revised version.
> > >
> > > 2. *Is there any recent papers like GFlowNet that could be referred other than synthesis-based SynNet published in 2022?*
> > >
> > > Thank you for the interesting suggestion! GFlowNet is indeed a related framework for molecular design. Its off-policy & offline learning setup would be applicable if we also queried the black-box oracle for rewards of our generated synthetic trees (which there are >100k of). In our case, the program synthesis policy learning is self-supervised, and its task is to infer valid programs for a given fingerprint, which is independent of properties. However, this can easily change if we introduce the notion of a reward function, and if so an amortized sampler like GFlowNet would indeed be useful for generating diverse and high-quality samples. We added a discussion of GFlowNet and Decision Transformer as alternate formulations in App. D.4.
> > >
> > > [1] Luo, Shitong, et al. "Projecting Molecules into Synthesizable Chemical Spaces." arXiv preprint arXiv:2406.04628 (2024).
> > >
> > > [2] Gao, Wenhao, Shitong Luo, and Connor W. Coley. "Generative Artificial Intelligence for Navigating Synthesizable Chemical Space." arXiv preprint arXiv:2410.03494 (2024).

---

> > > > ### Author Response · Authors · 2024-12-02
> > > > **Looking forward to your response**
> > > >
> > > > Dear reviwer,
> > > >
> > > > Thank you again for your valuable comments and thoughtful consideration! We’ve worked hard to thoroughly address your feedback and further refine the paper, aiming to make it the best it can be. We’d greatly appreciate your feedback on whether your concerns have been addressed!
> > > >
> > > > Sincerely,
> > > >
> > > > Authors

---

### Author Response · Authors · 2024-11-28
**Codebase and Rebuttals**

Dear reviewers,


Thank you for all the positive comments and constructive feedback. We’ve uploaded the revisions to our main text and supplementary information, with blue highlighting the changes we’ve made to the initial manuscript. Namely, we added two new ablation studies (4.1.3, 4.3.3), extensive descriptions and analyses of the ablation and results (App. B.1-B.3, App. F.5, App. F.7), added to existing sections (App. A), expanded related works (App. D.3), and other changes to the presentation of our paper (Introduction, 3.1, etc.) to incorporate the feedback you gave. We also release our anonymous GitHub here: https://anonymous.4open.science/r/SynthesisNet-6D4F/README.md.


We have also responded point-to-point to your concerns. Please let us know if there are any remaining points that need clarification before the discussion period closes. Finally, we encourage each reviewer to read the rebuttals given to other reviewers, as some points may have been addressed elsewhere.


Thanks and wishing you all happy Thanksgiving!

Authors

---

### Meta-Review · Area_Chair_kMsj · 2024-12-21

**Metareview:**

This paper casts molecule design and synthesis as a conditional program synthesis problem and then presents a framework for using program synthesis and machine learning techniques to accomplish the tasks of synthesizable analog recommendation and synthesizable molecule design. The experiments show improvements on certain metrics for both tasks.

**Additional Comments On Reviewer Discussion:**

The authors provided detailed responses to the concerns raised by the reviewers. There was a consensus among the reviewers that the paper should be accepted.

---

### Decision · Program_Chairs · 2025-01-22

Accept (Poster)